# Perylenetetracarboxylic acid nanosheets with internal electric fields and anisotropic charge migration for photocatalytic hydrogen evolution

Yan Guo[1,2], Qixin Zhou [2], Jun Nan[1], Wenxin Shi[3], Fuyi Cui[3] & Yongfa Zhu [2✉]

Highly efficient hydrogen evolution reactions carried out via photocatalysis using solar light remain a formidable challenge. Herein, perylenetetracarboxylic acid nanosheets with a monolayer thickness of ~1.5 nm were synthesized and shown to be active hydrogen evolution photocatalysts with production rates of 118.9 mmol g$^{-1}$ h$^{-1}$. The carboxyl groups increased the intensity of the internal electric fields of perylenetetracarboxylic acid from the perylene center to the carboxyl border by 10.3 times to promote charge-carrier separation. The photogenerated electrons and holes migrated to the edge and plane, respectively, to weaken charge-carrier recombination. Moreover, the perylenetetracarboxylic acid reduction potential increases from −0.47 V to −1.13 V due to the decreased molecular conjugation and enhances the reduction ability. In addition, the carboxyl groups created hydrophilic sites. This work provides a strategy to engineer the molecular structures of future efficient photocatalysts.

[1] School of Environment, Harbin Institute of Technology, 150090 Harbin, China. [2] Department of Chemistry, Tsinghua University, 100084 Beijing, China. [3] College of Environment and Ecology, Chongqing University, 400044 Chongqing, China. ✉email: Zhuyf@tsinghua.edu.cn

Semiconductor photocatalytic water splitting to produce $H_2$ is regarded as ideal to convert solar energy into clean hydrogen energy[1,2]. Many catalysts applied in the photocatalytic hydrogen evolution reaction (HER) are based on earth-abundant transition metals that are generally unstable in acidic HER environments[3]. Nevertheless, the *d* orbitals of transition metals[4] make it challenging to adjust the electronic structure[5]. In contrast, one key characteristic of organic photocatalysts with $\pi$ orbitals is their easily accessible electron structure tuning and processing flexibility into devices[6-8], leading to more suitability for the development of efficient photocatalysts. Stupp's group has carried out groundbreaking research on photocatalytic hydrogen production from perylenediimide-based self-assembled supramolecular materials[9-11]. Our group has made significant progress in exploring supramolecular photocatalyst structure-activity relationships[12-14]. The gradual improvement in photocatalytic performance has shown the potential of such materials[15]. Not only the HER but also most photocatalytic activity relies on thermodynamic conditions[16], charge separation[17], and surface reactions[18]. At present, most works focus on one of these aspects, but further commercial development of photocatalytic HER technology is still restricted due to its unsatisfactory activity[19]. Therefore, based on the engineering of organic molecular structures, this work aims to obtain an efficient HER by designing an organic photocatalyst integrating a high reduction potential, charge driving force and surface catalytic reaction.

Perylene plane series materials have become promising organic photocatalysts due to their $\pi$-$\pi$ stacking electron migration channels[20,21], cost-effectiveness, and broad light absorption[22], especially for low-energy solar energy[23]. However, these photocatalysts are not considered excellent photocatalytic HER candidates for the following reasons[9,24]. (1) Their reduction potential is insufficient to satisfy HER thermodynamic requirements[25,26]. (2) The lack of a driving force leads to the severe recombination of photogenerated carriers[27]. (3) Finally, a reduced number of hydrophilic sites results in a lower surface catalytic reaction. Considering the unique flexibility of organic photocatalysts, adjustment of the delocalized $\pi$-electrons to avoid the above has been envisaged[28-30]. This introduces appropriate groups to change the molecular energy level, such as electron-withdrawing groups lowering the frontier orbital level[31-34], further directly regulating the band structure of the semiconductor[35-37]. As a driving force, the semiconductor-enhanced internal electron fields (IEF) of semiconductors help to separate photogenerated carriers[17,38]. However, experimental evidence for the separation of carriers on different crystal planes remains elusive. Moreover, the surface catalytic reaction activity can be promoted by increasing the number of active sites[39,40]. Although the key bottlenecks and complementary methods are understood, integrating these strategies by regulating the molecular structures of photocatalysts remains challenging.

In this work, following the above design, supramolecular perylenetetracarboxylic acid (PTA) nanosheets were developed as photocatalysts for hydrogen evolution. To realize anisotropic separation of charge carriers, further avoiding recombination, an increased dipole from the perylene center to the carboxyl border of the PTA molecule was constructed. To obtain enhanced reduction potential for $H_2$ evolution, the conjugation effect of the molecule was properly attenuated. In addition, hydrophilic groups were employed to improve the surface-active sites for an efficient HER. By rational optimization of the dynamics, thermodynamic conditions, and surface reaction, the resultant PTA nanosheets showed enhanced photocatalytic HER activity that reached 118.9 mmol $g^{-1}$ $h^{-1}$. Moreover, PTA has application value because it is flexible to process into devices.

## Results and discussion

**PTA nanosheets photocatalyst structure.** We synthesized PTA nanosheets by the facile hydrolysis-reassembly of PTCDA (Methods). The chemical structure of the synthesized PTA was first confirmed. The[13]C ssNMR spectrum revealed five chemical environments of carbon in PTA (Supplementary Fig. 1a). The time-of-flight secondary ion mass spectrometry (TOF-SIMS) was performed to reconfirm the supramolecular PTA. As shown in Fig. 1a, a prominent molecular ion peak corresponding to PTA appears at a mass-to-charge ratio (m/z) of 428.05. The ion peak of the dehydroxylated fragments ($C_{24}H_{11}O_7$ fragments) resulted from sputtering ionization observed at m/z = 411.04. In addition, Fourier transform infrared (FT-IR) spectroscopy, X-ray photoelectron spectroscopy (XPS) and energy dispersive spectrometry afforded more detailed information regarding the characteristic structure of PTA (Supplementary Figs. 1b-c, 2–4). The above results identified the accuracy of the PTA structure.

The PTA nanosheets were formed by molecules self-assembled via $\pi$-$\pi$ stacking between the perylene plane. X-ray diffraction (XRD) in Fig. 1b investigates the crystal structure. The peak at $2\theta = 22.7°$ can be assigned to the $\pi$-$\pi$ stacking between the perylene plane with a distance of 3.9 Å referring to the (021) facet. This $\pi$-$\pi$ stacking provides electron migration channels from the bulk to the surface[13]. Moreover, the $\pi$-$\pi$ channels enhance the excitonic coupling of the system, allowing the PTA nanosheets to exhibit a fluorescence quantum yield of 4.9% in water (Supplementary Note 1)[9]. The electron diffraction reconfirmed this crystal feature, further inferring that the (100) facet is the exposed surface of PTA (inset in Fig. 1c). The transmission electron microscopy (TEM) images revealed that PTA consists of stacked nanosheets with abundant overlapping areas (Fig. 1c). The detailed morphology of PTA was studied by field emission scanning electron microscopy (FESEM, Supplementary Fig. 7). The ~1.5 nm thickness predominated by PTA nanosheets was observed by atomic force microscopy (AFM), and ~3 nm corresponds to two layers (Fig. 1d). As shown in Fig. 1e, the thickness refers to the length of the PTA molecule, indicating that a 1.5 nm-thick sheet is composed of PTA molecules via $\pi$-$\pi$ interactions. Notably, given the correspondence between the height of PTA nanosheets and the length of a PTA molecule, the (100) facet is reconfirmed as the exposed identical facets. Since the migration distance of excitons is <10 nm[41-43], PTA with an ultrathin crystal is beneficial to reduce the recombination of photogenerated carriers during migration.

Furthermore, the nanosheets morphology is thermodynamically stable (Supplementary Note 7). Although hydrogen bonding exists between the carboxyl groups and $H_2O$ molecules, the nanosheets morphology persists in the water environment due to $\pi$-$\pi$ interactions between PTA molecules and the protection of the hydration layer. Theoretical calculations and experimental characterization provide evidence (Supplementary Figs. 8–14, & Supplementary Table 1).

**Photocatalytic hydrogen evolution.** PTA nanosheets exhibited excellent photocatalytic HER performance under the full spectrum with the aid of a Pt cocatalyst (the photon flux was ~530 mW cm$^{-2}$ provided by Xe light of $\lambda \geq 300$ nm). The optimum dosage of photocatalyst and Pt cocatalyst was 7 mg and 4.6 wt. % (Supplementary Fig. 15), respectively. As shown in Fig. 2a, the average HER rate of PTA occurring at the optimal dosage reached 118.9 mmol $g^{-1}$ $h^{-1}$ (832.3 µmol $h^{-1}$). The value was 81.6 mmol $g^{-1}$ $h^{-1}$ (571.2 µmol $h^{-1}$) under visible light ($\lambda \geq 420$ nm) irradiation. Under AM 1.5 G (100 mW cm$^{-2}$) simulated sunlight, the hydrogen evolution from PTA nanosheets reached 41.8 mmol $g^{-1}$ $h^{-1}$ (292.6 µmol $h^{-1}$). This is one of

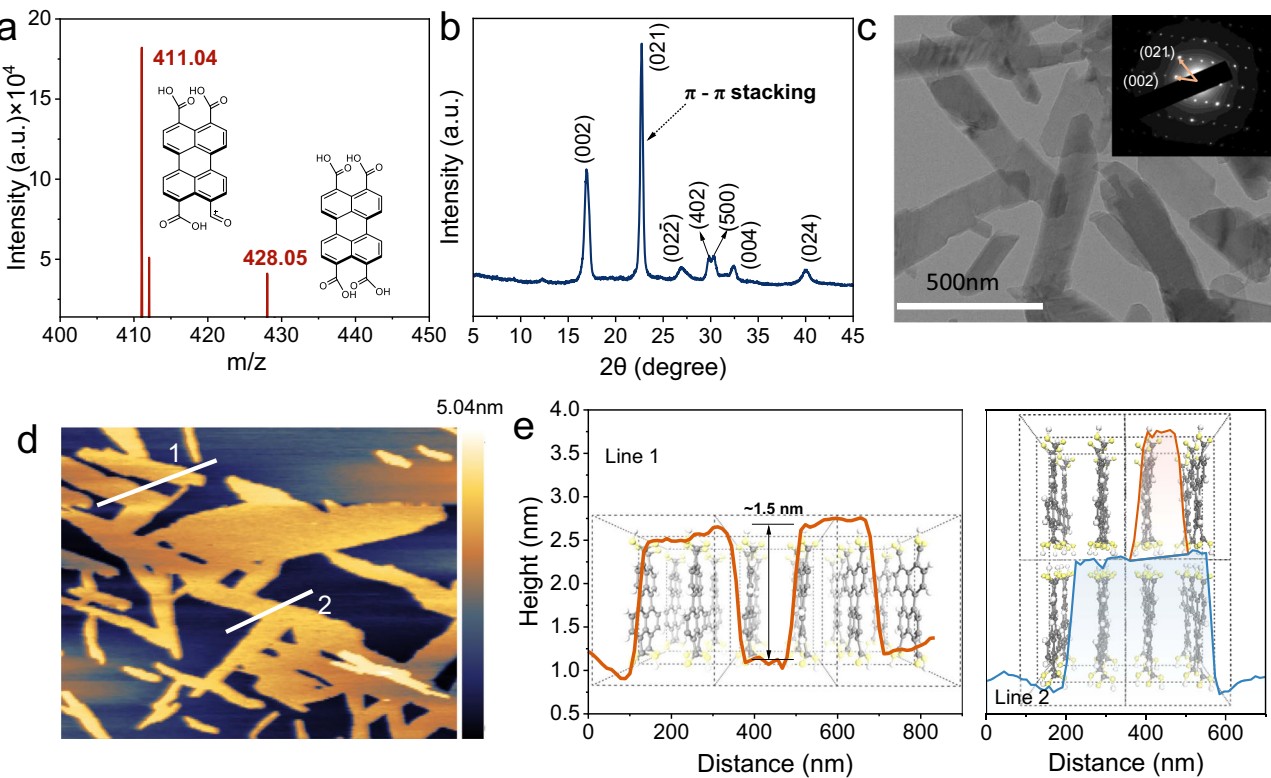

**Fig. 1 Structural characterization of PTA nanosheets. a** Time-of-flight secondary ion mass spectrometry (TOF-SIMS), **b** X-ray diffraction (XRD), **c** transmission electron microscope (TEM), inset: electron diffraction pattern, **d** atomic force microscope (AFM) image, and **e** corresponding height and simulated structure of PTA nanosheets.

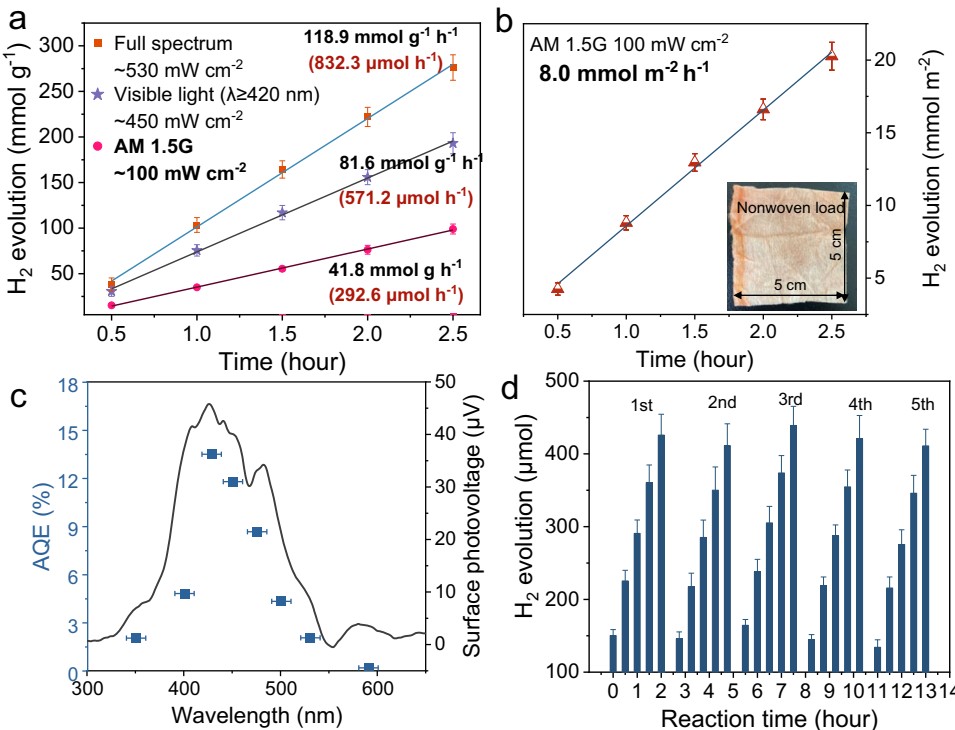

**Fig. 2 Photocatalytic hydrogen evolution. a** Photocatalytic HER under full spectrum (λ ≥ 300 nm, light intensity: ~530 mW cm⁻²), visible light (λ ≥ 420 nm, light intensity: ~450 mW cm⁻²) and AM 1.5 G, (100 mW cm⁻²) of PTA nanosheets. **b** AM 1.5 G (100 mW cm⁻²) HER performance of PTA (2 mg) loaded on nonwoven fabrics, inset: optical image of PTA loaded on nonwoven devices. (All error bars are determined from three independent experiments). **c** The wavelength-dependent AQE for photocatalytic HER over PTA and the surface photovoltage of PTA. (The error bar came from a bandpass filter). **d** The cycle photocatalytic HER performance of PTA under the full spectrum. Reaction conditions: 7.0 mg photocatalyst, 4.6 wt. % Pt as a cocatalyst, 100 mL of a 0.2 M ascorbic acid solution, pH 2.45. (All error bars were determined from three independent experiments).

the best organic photocatalysts reported thus far, surpassing most metal-containing inorganic photocatalysts (Supplementary Table 3). However, the $H_2$ evolution of the original perylene series PTCDA is lower (6.9 mmol $g^{-1} h^{-1}$ under $\lambda \geq 300$ nm and 1.86 mmol $g^{-1} h^{-1}$ under $\lambda \geq 420$ nm) than that of PTA (Supplementary Fig. 17a).

Benefiting from the flexibility of organic materials[44], PTA was sprayed on hydrophilic nonwoven fabrics to form a solar HER device (Methods). The HER performance of the device reached 8.0 mmol $m^{-2} h^{-1}$ under AM 1.5 G. Assuming 8 h of irradiation per day, it can produce ~1.4 L of hydrogen per square meter per day (Fig. 2b). The wavelength-dependent HER performance exhibits a broad wavelength response of PTA, showing typical photocatalytic behavior (Supplementary Fig. 17b). Photocatalytic reactions occur on the surface, indicating that the surface charge dominates the HER performance. The surface photovoltage (SPV) spectrum was employed to detect the photoelectric response on the PTA surface. A significant signal is produced at 300–600 nm and showed a maximum of 45.7 µV at 425 nm. The apparent quantum efficiency (AQE) was measured at different light wavelengths by monochromatic light (Supplementary Table 4). The wavelength-dependent AQE trend follows that of SPV, indicating bandgap-transition-dependent hydrogen production[45]. In addition, the AQE value at 420 nm (±10 nm) was calculated to be 13.5% (Fig. 2c), showing the best hydrogen quantum efficiency among similar photocatalytic materials.

Furthermore, the HER activity could be retained to exceed five individual cycles (Fig. 2d) and the continuous hydrogen production over 24 h without a significant decrease under AM 1.5 G (Supplementary Fig. 18), displaying the stability of the PTA nanosheets. The morphology and chemical structure of PTA were maintained after recycling (Supplementary Figs. 19–20). The stable $\pi$-$\pi$ stacking of the PTA nanosheets ensures stability (Supplementary Note 8). Inspired by the efficient photocatalytic HER performance of PTA, the role of the PTA molecular structure in thermodynamic and dynamic conditions and surface reactions was examined.

**Reduction potential**. The semiconductor band position is a thermodynamic decisive component of the photocatalytic reduction reaction[46]. The degeneracy of isolated molecular energy levels widens to form degenerate conduction bands (CBs) of organic crystals. The steric hindrance of the carboxylic acid and the distortion of the perylene plane reduce molecular coplanarity, leading to a reduced conjugation effect across the whole molecule (Supplementary Fig. 21). In detail, $\pi$ orbital dispersion in molecules exhibits the degree of conjugation effect[47]. The $\pi$ orbital of

the PTCDA molecule is extended to the entire molecular plane, as presented by the localized orbital locator-$\pi$ (LOL-$\pi$) isosurface (Fig. 3a). However, a nodal plane exists on the $\pi$ orbital of the PTA molecule where the perylene plane intersects with the carboxyl group. This decreased molecular conjugation effect causes a wider bandgap, which was reconfirmed by the calculated molecular orbitals (Supplementary Figs. 22–23). Furthermore, the band structure formed by the energy level degeneracy exhibited the same trend. Both the optical band gap predicted by the SPV (Supplementary Fig. 24) and the band structure calculated (Supplementary Figs. 25–26) proved this. Note that the CB is more affected by the bandgap than the valence band, elevating the CB position.

Then, the band positions of the semiconductors were tested experimentally. According to the Mott-Schottky curve[48], the CB potential of PTA (−1.13 V vs. SHE, pH = 7) was higher than that of PTCDA (−0.47 V vs. SHE, pH = 7) (Supplementary Fig. 24). Knowing the bandgap values and the CB position, the band positions of PTCDA and PTA are outlined in Fig. 3b. Although both are thermodynamically favorable for the HER (−0.41 V vs. SHE, pH = 7), a reduction potential that is much higher than the demand is undoubtedly advantageous[49,50]. The above results indicate that the carboxyl groups decrease the molecular conjugation effect and increase the CB position, endowing enough reduction potential for PTA nanosheets.

**Charge separation and anisotropic migration**. As a driving force for the separation and migration of photogenerated charge[51], IEF is a dynamic factor affecting photocatalytic performance[52–54]. The molecular dipole moments are from the center to the edge in PTCDA and PTA, 1.43 and 4.98 Debye, respectively. The value of PTA is 3.5 times stronger than that of PTCDA with the assistance of marginal carboxyl groups (Supplementary Fig. 27). The electron excitation differential density further confirmed this. As shown in Fig. 4a, the electron cloud trended to be enriched in carbonyl oxygen compared with the perylene plane in PTA. The excited charge density difference was $4.53 \times 10^{-3}$ eV $Å^{-3}$. PTCDA has a similar tendency but is weaker ($0.97 \times 10^{-3}$ eV $Å^{-3}$) than PTA. The theoretical results imply that the enhanced IEF exists in the PTA molecule from the perylene plane to the carboxyl polar group on the edge. In addition, the hydrogen bond polarization of carboxyl groups in PTA can strengthen the IEF[55]. The ordered crystallinity of PTA contributes to the accumulation of intramolecular dipole-induced electric fields in the macroscopic IEF[56] (Supplementary Fig. 28). The Poisson equation describes the spatial distribution of IEF, which shows that IEF is a function of surface potential and surface charge density[57–59] (Supplementary

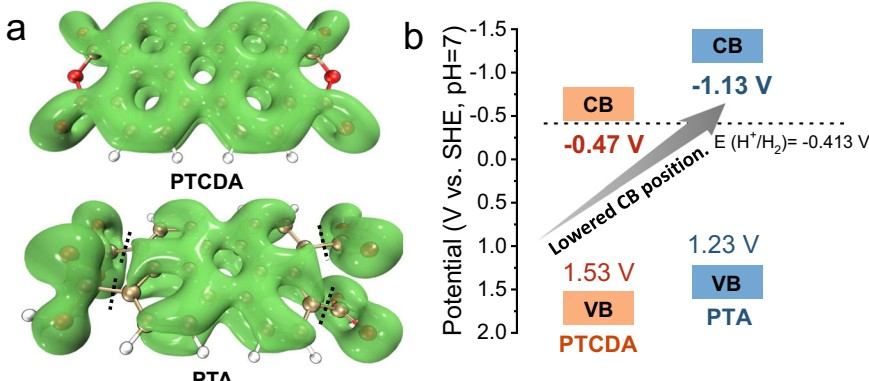

**Fig. 3 The reduction potential of the PTA photocatalyst. a** The localized orbital locator-$\pi$ (LOL-$\pi$) isosurface. **b** The band structures of PTCDA and PTA measured experimentally.

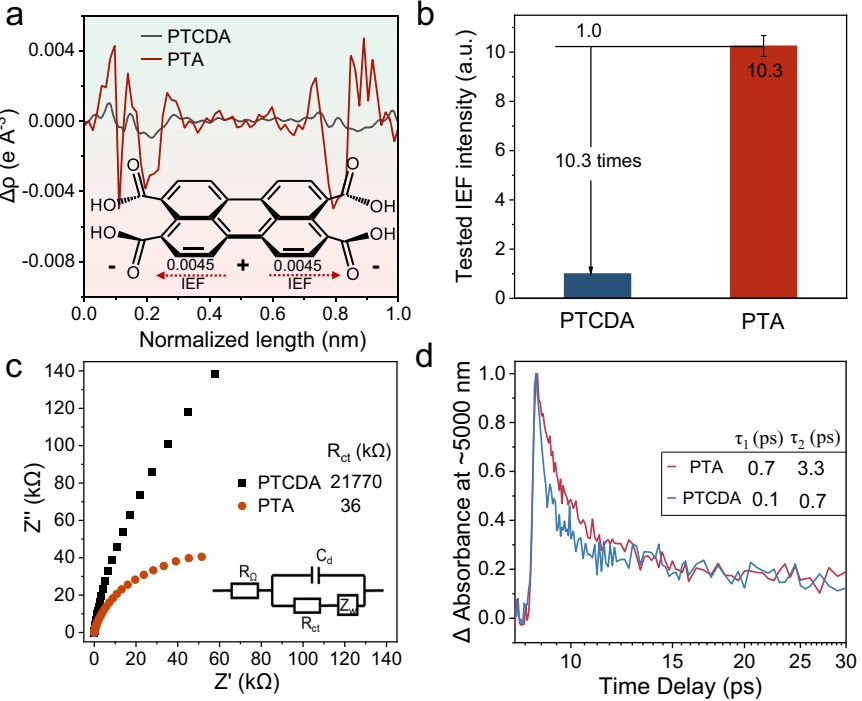

**Fig. 4 IEF drives charge separation. a** Excited charge density difference between the PTA and PTCDA cells (set with carbonyl oxygen). **b** The tested IEF intensity, **c** impedance and **d** transient absorption decay kinetics at ~5000 nm of PTCDA and PTA.

Methods 2). AFM detects the surface potential with a Kelvin probe[60–63]. The potential difference between the surface of PTCDA and the substrate (HOPG) was only 27 mV, while that of PTA was 70 mV (Supplementary Fig. 29). The surface charge density is a function of zeta potential[64,65], and the zeta potential of PTA ($\zeta = \sim -13.63$ mV) is significantly higher than that of PTCDA ($\zeta = \sim -3.93$ mV) (Supplementary Fig. 30). According to the normalization of the calculated results, the IEF of PTA is 10.3 times that of PTCDA (Fig. 4b, Supplementary Methods 2).

The long-range ordered nanosheets structure weakens the exciton binding energy of PTA, while excitons dissociate more readily into photogenerated carriers under the enhanced IEF effect (Supplementary Note 2). This is evidenced by the weak charge migration resistance in the semiconductor. The impedance values (Fig. 4c) of PTA[66–68] (the equivalent circuit fitting resistance is 36 kΩ) were less than those of PTCDA (21,770 kΩ). As a result, the increased electron mobility (Supplementary Note 3) is 2.9-fold, illustrating that the enhanced IEF contributes to charge migration from the bulk to the surface. With such an IEF, the carrier generation rate of the PTA nanosheets reached 90.6% (Supplementary Note 4). Furthermore, the photocurrent displays numerous photogenerated charges reaching the surface (Supplementary Fig. 31).

We probed the charge decay kinetics by femtosecond-resolved transient absorption spectroscopy (TAS)[69,70]. The positive absorbance represents the excited state absorption (ESA) of S1-Sn (Supplementary Fig. 32)[71,72]. The ESA signals representing photogenerated holes (at ~650 nm) and electrons (at ~5000 nm) are assigned in the presence of holes and electron scavengers, respectively (Supplementary Figs. 33–35). The measured changes in the mid-infrared (MIR) region were mainly caused by the absorption of photogenerated carriers[73,74]. We fitted the decay kinetics of photogenerated carriers using a second-order exponential fit, where $\tau_1$ indicates that the photogenerated carrier is rapidly compounded after exciton dissociation and $\tau_2$ indicates that the photogenerated carrier is trapped during migration after

dissociation (Supplementary Note 5). Compared to PTCDA, IEF inhibits the recombination of photogenerated carriers (presented as the prolongation of $\tau_1$). The $\tau_2$ values for the photogenerated holes (at ~650 nm) and photogenerated electrons (at ~5000 nm) of PTA were prolonged by 3.2- (Supplementary Fig. 36) and 4.7-fold (Fig. 4d), respectively. Excluding the effect of morphology on photogenerated carrier behavior, the longer decay is attributed to the IEF driving force increasing the surviving photogenerated carriers (Supplementary Note 6).

We next sought to determine the orientation of the IEF and the effect on anisotropic separation of photogenerated carriers by photodeposition experiments. With $NaIO_3$ as the electron scavenger, $Mn^{2+}$ can be oxidized to $Mn_2O_3$ oxide by photogenerated holes. Taking ascorbic acid as the hole scavenger, the metal ion ($Pt^{4+}$) is reduced to Pt by photogenerated electrons. Thus, the deposition sites of $Mn_2O_3$ and Pt can be used to explore the photogenerated holes and electron generation sites of PTA nanosheets. XPS of Pt 4*f* revealed that the deposited elements are metallic (Supplementary Fig. 38a). The distance between the two peaks of Mn 3*s* is 5.35 eV (Supplementary Fig. 38b), ascribing the deposited manganese species to $Mn_2O_3$[75–77]. TEM (Supplementary Fig. 39a) and high-angle annular dark-field STEM (HAADF-STEM) images (Fig. 5a) clearly showed that the Pt particles were almost deposited on the edge of PTA (that is, the (010) facet). The 2.2 Å representing Pt (111) facets was observed in the high-resolution TEM images. The formation of Pt at the edge confirmed that the photogenerated electron tended to migrate to the (010) facet for metal reduction. As expected, $Mn_2O_3$ was deposited on the plane (that is, (100) facet) of PTA (Supplementary Fig. 40), which was further verified by a line scan (Fig. 5b). The formation of $Mn_2O_3$ on the plane suggested that photogenerated holes tend to migrate to (100) facets. Notably, the excitons do not move under the electric field[78]; thus, the directional deposition of Pt and $Mn_2O_3$ is induced by the carriers in the dissociated state. The model shown in Fig. 5c describes the IEF-driven photogenerated carrier migration

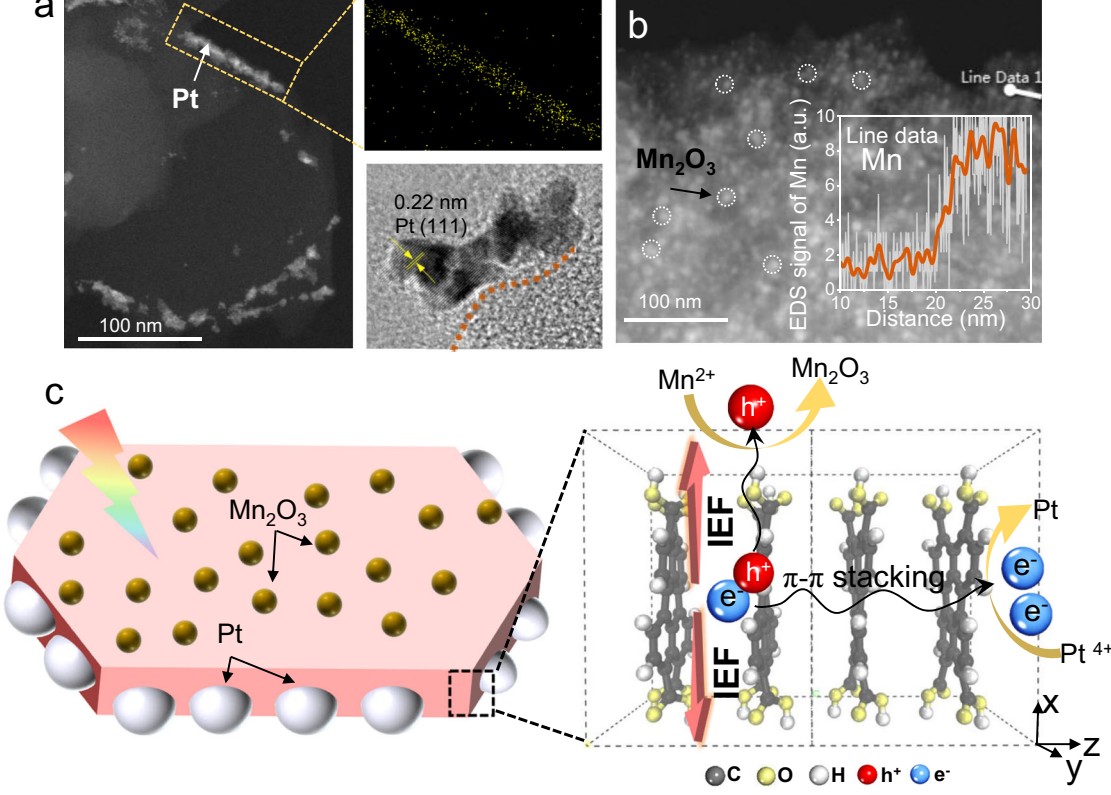

**Fig. 5 Anisotropic migration of photogenerated carriers.** ADF-STEM photodeposition on PTA with **a** Pt and high resolution and mapping, **b** $Mn_2O_3$ (inset: Mn element liner scan). **c** Schematic illustration of Pt and $Mn_2O_3$ photodeposition on PTA under visible light ($\lambda \geq 420$ nm). Deposition method: 10 mg of PTA was dispersed in 100 mL of deionized water, and 4 wt. % $H_2PtCl_6$ (weighed by Pt) or 5 wt. % $MnCl_2$ (weighed by Mn) was added and exposed to visible light ($\lambda \geq 420$ nm) for 3 h.

toward the (010) and (100) facets. The IEF enhanced by polar groups and crystallinity produces anisotropic charge migration to promote HER activity.

**Hydrophilic sites for hydrogen evolution.** Full contact between the photocatalyst and water benefits efficient surface reaction activity[79,80]. We investigated the role of the carboxyl group in the water adsorption capacity by the water contact angle. After 1 s of contact with water droplets, the average contact angle of PTA was only 30.5°, while that of PTCDA was 59.7° (Fig. 6a). Moreover, the PTA nanosheets absorbed the water droplets entirely after ~2 s. The polar carboxyl groups in the molecule endow the PTA nanosheets with hydrophilicity and dispersibility in water. The sufficient water-photocatalyst contact suggests a high surface reaction probability[81]. A careful examination by ATR-FTIR measurement with the changes of photocatalyst adsorbed isotope-labeled $D_2O$ to demonstrate the hydrophilic water sites. The vibration peaks related to carboxyl groups moved to low wavenumbers regularly after $D_2O$ adsorption, in line with the vibration frequency ratio of valence bonds after isotope substitution (O–H: O–D = ~0.7). Moreover, the formation of $H_2O$ was observed at ~3300 $cm^{-1}$, which proved that H/D exchange occurred with the carboxyl group (Fig. 6b). However, there was no change in the FTIR spectrum after $D_2O$ adsorption for PTCDA, indicating the few hydrophilic sites of PTCDA (Supplementary Fig. 41). The above evidence shows that PTA has widely distributed hydrophilic sites originating from the carboxyl group.

To gain insight into the source of protons reduced by PTA, $D_2O$ was used instead of $H_2O$ during the photocatalytic HER.

The rate of $D_2$ evolution can reach 33.95 mmol $g^{-1} h^{-1}$ under the full spectrum (Fig. 6c), indicating that the origin of $H_2$ is aqueous protons[82]. The presence of the H/D isotope effect decreases the photocatalytic HER[83], but in principle, it does not harm our inferences about the source of $H_2$ in the reaction systems. To rule out the possibility that the protons were from hole scavengers (ascorbic acid, AA), the high-resolution mass spectra of the solution before and after the HER were applied. After the reaction, a prominent peak of m/z = 191.02 for $C_6H_8O_7$ (2,3-diketo-L-gulonic acid, oxidization of AA) appeared (Supplementary Figs. 42–43). No peaks of $C_6H_6O_6$ corresponding to the dehydrogenated AA products were detected. $C_6H_8O_7$ is produced from the reaction of $C_6H_8O_6$ with a photogenerated hole ($h^+$), which gives definitive evidence that AA serves as a hole scavenger instead of the source of hydrogen. These results rule out the debate about the HER proton source, further confirming the importance of hydrophilic sites. Thus, an effective perylene series HER photocatalyst was constructed with the assistance of carboxyl groups, which could significantly increase the reduction potential, decrease carrier recombination and provide hydrophilic sites. These factors contributed to the efficient photocatalytic HER process, and we summarized the working mechanism in Fig. 7.

In summary, we demonstrated the enhanced HER performance of visible light-responsive PTA nanosheets as an advanced and practical photocatalyst. The anisotropic charge migration via IEF to avoid recombination plays a critical role in efficient $H_2$ evolution (118.9 mmol $g^{-1} h^{-1}$). Moreover, the decreased conjugation effect in PTA enhances the reduction potential for $H_2$ evolution. In addition, the abundant hydrophilic sites formed by the carboxyl

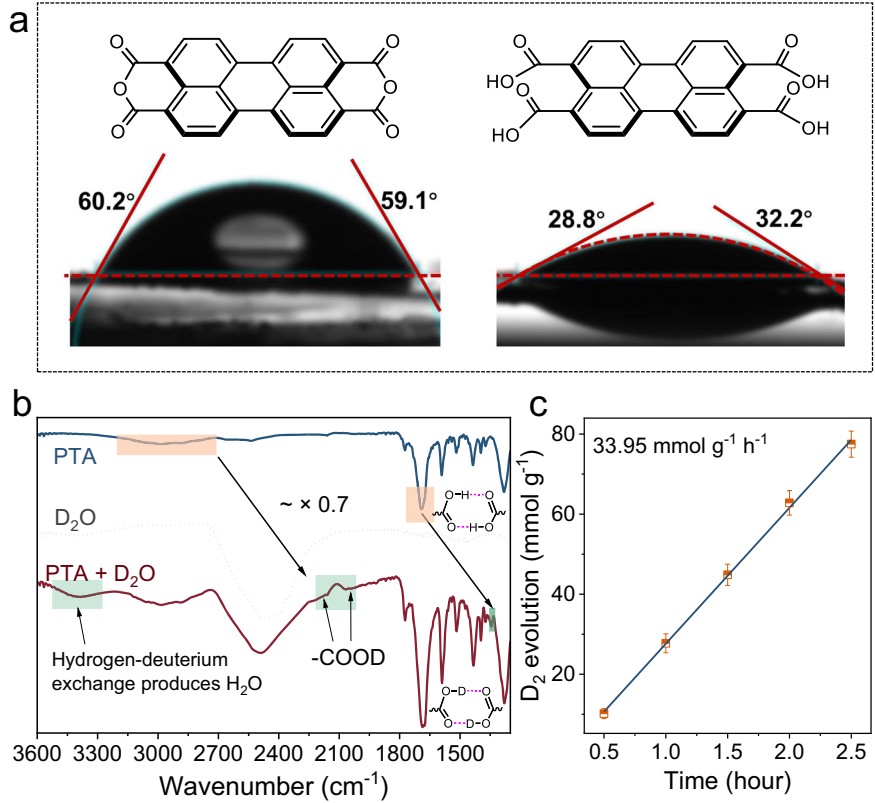

**Fig. 6 Hydrophilic sites for hydrogen evolution. a** Contact angle of PTCDA and PTA with water. **b** ATR-FTIR spectra of PTA before and after in situ adsorption of $D_2O$. **c** $D_2$ evolution performance of PTA-split $D_2O$.

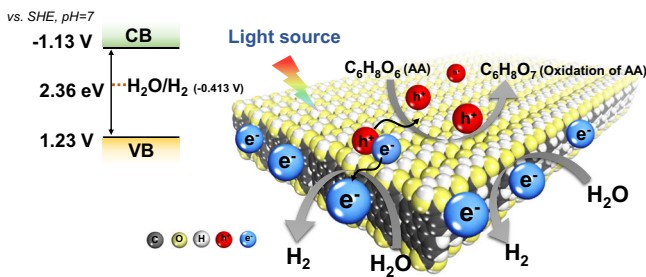

**Fig. 7 Schematic mechanism of the photocatalytic HER by PTA nanosheets.** Left: Band positions of PTA nanosheets. Right: Migration and reaction of photogenerated charges during the photocatalytic HER of PTA nanosheets.

groups assist with the high activity. We revealed the mechanism of IEF formation and driving charge anisotropic migration in supramolecular nanosheets. Since organic semiconductors have flexible tunable structures, this strategy is expected to extend various artificial photosynthesis applications.

## Methods

### Materials

*All chemicals used in the research were purchased commercially without additional purification. Preparation of perylene-3,4,9,10-tetracarboxylic acid (PTA).* PTCDA was purchased from Sigma. Fifty milligrams of PTCDA was dispersed in 150 mL of deionized water. After sonicating for 20 min, 0.1 g of KOH was added and quickly placed in a water bath at ~50 °C. The mixture was stirred for 0.5 h to form yellow–green transparent solution A. A total of 0.5 mL of acetic acid with deionized water was diluted to 50 mL, called solution B. After waiting for solution A to cool to room temperature, the mixture was poured into solution B under rapid stirring. After stirring for 2 h, the suspension was washed with suction to neutrality by deionized water, and the precipitate was collected and dried at 60 °C. The formation of PTA is a self-assembly process regulated by π–π

interactions, hydrogen bonds and hydrophilic and hydrophobic interactions (Supplementary Fig. 44).

PTA loaded on nonwoven fabrics: PTA (2 mg) was dispersed in 10 mL deionized water. Two milliliters of evenly dispersed droplets were spread on clean and dry nonwoven fabric and further dried at 60 °C for 3 h. This process was repeated until all drops were applied to the nonwoven fabric.

**Sample preparation method for the TAS test.** TAS for testing solid coating films: 1 mg of the sample was dispersed in 2 mL of deionized water and sonicated to disperse well. The substrate was coated three times, and each coating was dried at 50 °C before the next drop. To test the TAS of PTA with scavengers of photo-generated electrons and holes, 0.2 M ascorbic acid or $AgNO_3$ was used as the scavenger to replace the above deionized water. For samples measured in the visible region, quartz glass was used as a substrate. For samples measured in the MIR region, 0.5 mm thick $CaF_2$ plates were used as substrates.

Testing the TAS of samples dispersed in water: $1 \times 10^{-5}$ M PTA was placed in water, dispersed well by sonication and placed in a cuvette to be measured. See Supplementary Figs. 32–36 for the TAS involved in this study.

### Characterization

*General.* The xenon lamp's intensity was tested with an optical power meter (Thorlabs PM100D). Morphology was studied by TEM with a Hitachi HT7700 (the accelerating voltage was 100 kV). Further morphology was explored using a field emission scanning electron microscopy (FESEM, model: Hitachi SU-8010). The XRD patterns of the samples were measured on a Rigaku D/max-2400 X-ray diffractometer using Cu Kα1 radiation (λ = 0.15418 nm) at 45 kV and 200 mA, with a scan step of 5°. The Anton Paar, Austria SAXS ess mc2 was applied to test the small angle X-ray scattering (SAXS) with a wavelength of 0.15406 nm, voltage of 40 kV, current of 30 mA and point light source. Fourier transform infrared (FT-IR) spectra were acquired on a Bruker V70 spectrometer. A Spm-9600 made by Shimadzu and Oxford in Japan Cypher VRS was used to test the sample height by using mica sheets as a substrate. The surface potential of the samples was tested on a substrate of high directional pyrolytic graphite. An Edinburgh F900 fluorescence spectrophotometer was used to analyze the steady-state photoluminescence (PL) spectra. The SPV spectroscopy detection instrument was composed of an optical chopper (SR540, USA) synchronized with a lock-in amplifier (SR830, USA).

The zeta potential of the photocatalyst was measured with a SZ100 particle size analyzer produced by HORIBA in Japan and a zeta potential analyzer. A Jnm-ecz600r solid-state nuclear magnetic resonance apparatus was employed to test the

bonding information of carbon. TOF.SIMS 5–100 (ION-TOF GmbH) was used to analyze the relative molecular mass of supramolecular surface monomers and their fragments. Primary ion beam: $Bi^{3+}$, 30 keV, 45° incident, scanning area: 200 μm × 200 μm, secondary ion polarity and mass range: positive and negative ions, 0–33,600 amu, sputtering ion beam: GCIB, 10 keV, 45° incidence, sputtering rate = 0.18 nm $s^{-1}$ for $SiO_2$. The data from positive ion mode after sputtering were used in this paper. A MALDI-TOF/TOF mass spectrometer (Axima-Performancema) was used to acquire the mass spectra before and after the ascorbic acid reaction. The electron mobility was estimated by the space-charge-limited current method with a source meter (Keithley 2400). Electron paramagnetic resonance (EPR) was examined with a Bruck EMX-Plus.

*Transient absorption*. The laser source uses a femtosecond amplified laser system (35 fs, 1 kHz, 800 nm, Spitfire Ace, Spectra Physics). There are two output pulses. The first generates a second harmonic by a nonlinear optical crystal (α-BBO) to produce a 400 nm laser pulse. The second is focused on 4 mm-thick $CaF_2$ as a probe. A laser frequency synchronized fiber optic spectrometer (AvaSpec-ULS2048CL-EVO, Avantes) acquired the transmitted probe light after the sample. The pumping laser beam was cut at a laser frequency of 1 kHz to obtain the probe light. Subsequently, the change in sample optical density due to the pumping pulse was obtained. A pumping intensity of 5 μJ/$cm^2$ was used in this work.

Specifically, the transient absorption spectrum in the MIR region is presented. The Ti:Sapphire laser (Spitfire Ace, Spectra Physics) outputs an 800 nm central pulsed laser (35 fs, 1 kHz) splitting into two beams via a beam splitter. One beam acts as an excitation beam by passing through a delay line. For the other beam, four-wave mixing of the filament in air produces a broadband MIR source that serves as a MIR probe. The time delay of the fundamental beam is fine-tuned using a short delay line, and then the polarization of its light is rotated by 90° using a half-wave plate and finally recombined with the second harmonic beam through a dichroic mirror. The transmitted MIR light was collected into an imaging spectrometer (iHR 320, HORIBA Jobin Yvon), whose optical signal was received using a 64-channel MCT array detector of a femtosecond pulse acquisition technique (FPAS-0144, Infrared Systems Development). The excitation wavelength was 400 nm with an excitation energy of 5 μJ/$cm^2$.

**Photoelectrochemical measurements**. The photoelectrochemical properties were measured on a CHI-660 E electrochemical workstation. In a three-electrode cell, Ag/AgCl was used as the reference electrode, Pt wire was used as the counter electrode, and the prepared sample was covered with ITO glass as the working electrode. The electrolyte solution was 0.1 M $Na_2SO_4$. Working electrode preparation method: A total of 1 mg of sample was weighed and dispersed in water and further ultrasonicated to make the dispersion uniform. Finally, it was drop-coated on ITO glass and dried naturally (Supplementary Fig. 45). Electro-chemical impedance experiments: EIS spectra were obtained by perturbing the system with an AC signal of 1000–10 kHz at an amplitude of 10 mV. The equivalent circuit method was used to resolve the charge transfer resistance of the catalyst.

The conversion of the Ag/AgCl reference potential to a reversible hydrogen electrode (SHE) was calculated using Eq. 1:

$$E(SHE) = E(Ag/AgCl) + E^0(Ag/AgCl) + 0.059 \times pH \quad (1)$$

where $E^0(Ag/AgCl)$ is 0.197 V, and the pH of the electrolyte is nearly 7.

**Photocatalytic hydrogen evolution experiments**. The photocatalytic HER was carried out in a top-irradiated reaction vessel at a fixed temperature of 25 °C and connected to a glass-enclosed gas system (Labsolar-6A, Beijing Perfect Light Source). In a typical run, 4.6 wt. % Pt was photodeposited in situ onto a photo-catalyst as a cocatalyst (with $H_2PtCl_6$ as the Pt source). The reaction system (pH = 2.45) consisted of 100 mL of deionized water, 7 mg of photocatalyst, and 0.2 M ascorbic acid (photogenerated hole scavenger). The suspension and the glass system were thoroughly evacuated before the reaction started. Light source and light intensity: A 300 W Xe lamp (~530 mW $cm^{-2}$) provided the full spectrum, a cutoff filter (λ ≥ 420 nm, ~450 mW $cm^{-2}$) equipped with visible light, or AM 1.5 G (100 mW $cm^{-2}$) simulated sunlight. The gas evolved in the reactor was analyzed by an online gas chromatograph (GC-2002 N/TFF) equipped with a 5 Å molecular sieve column and Ar as the carrier gas TCD detector.

The AQE, or sometimes the external quantum efficiency, can be described by Eq. 2. The irradiation area was controlled at $1.0 \times 1.0$ $cm^2$.

$$AQE(\%) = \frac{\text{the number of evolved } H_2 \text{ molecules} \times 2}{\text{number of incident photons}} \times 100\% \quad (2)$$

**Photodeposition experiments**. Deposition method: First, 10 mg of PTA was dispersed in 100 mL of deionized water, and 4 wt. % $H_2PtCl_6$ (weighed by Pt) or 5 wt. % $MnCl_2$ (weighed by Mn) was added and exposed to visible light (λ ≥ 420 nm) for 3 h.

A decrease in the valence states of the metal ions ($Pt^{4+}$) indicates that the metal ions are reduced by the photogenerated electrons of the photocatalyst, and the

photogenerated holes are eliminated using ascorbic acid as a scavenger. The reaction can be described as Eqs. 3 and 4:

$$Pt^{4+} + e^- \rightarrow Pt \quad (3)$$

$$C_6H_7O_6^- + h^+ \rightarrow C_6H_6O_6^{2-} + H^+ \quad (4)$$

With $NaIO_3$ as the electron scavenger, $MnCl_2$ can be oxidized to $Mn_2O_3$ oxide under the oxidation of photogenerated holes. The reaction can be summarized as Eqs. 5 and 6:

$$Mn^{2+} + H_2O + h^+ \rightarrow Mn_2O_3 + H^+ \quad (5)$$

$$IO_3^- + e^- \rightarrow I^- + O_2 \quad (6)$$

**Computational methods**. Implement the Perdew-Wang exchange correlation function (PW91) with the CASTEP code. The charge density difference of PTA was calculated by the generalized gradient approximation. The hybrid Heyd-Scuseria-Ernzerhof 03 (HSE 03) functional with distance separation was applied in the band gap calculation. The cutoff energy of the plane wavefunction is 500 eV. The periodically replicated slabs were separated by a vacuum region of ~15 Å. A Monkhorst-Pack grid of 6*8*6 was used.

Computational investigations of the interactions between multiple molecules and water were performed using the ab initio quantum chemistry calculation method implemented by the VAMP simulation package with the PM6 semiempirical method.

## Data availability
Source data are provided on this platform (https://doi.org/10.5281/zenodo.6404530). Source data are provided with this paper.

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

## Acknowledgements

This work was supported by the National Science Foundation of China (21872077), the National Key Research and Development Project of China (2020YFA0710304), and the Collaborative Innovation Center for Regional Environmental Quality. We thank Dr. Ruilong Zong and Chao Ma at the Analysis Center of Tsinghua University for their help with TEM. We thank Shufen Yue at the Analysis Center of Tsinghua University for her help with AFM detection. We thank Dr. Haifang Li at the Analysis Center of Tsinghua University for her help with MS detection. We thank Dr. Wenzhe Li of Jinan University for his help with electron mobility testing.

## Author contributions

Y.G. performed most of the experiments and wrote the first version of the paper. J.N. revised the paper. Q.X.Z. carried out the DFT calculations and helped to analyze the results. Y.F.Z. supervised and took part in the revision of the paper. W.X.S. and F.Y.C. participated in the revision. All authors discussed the results and commented on the paper.

## Competing interests

The authors declare no competing interests.
