## [Peer Review File · Nature Communications]

Perylenetetracarboxylic Acid Nanosheets with Internal Electric Fields and Anisotropic Charge Migration for Photocatalytic Hydrogen EvolutionREVIEWER COMMENTS

Reviewer #1 (Remarks to the Author):

The manuscript by Guo et al. details the synthesis of perylene tetracarboxylic acid (PTA) aggregates and their structural, photophysical, and photocatalytic behavior. The authors find that the PTA forms nanosheet structures with nanolayer thickness. Upon irradiation, hydrogen evolution is seen at a high rate of 119 mmol/g/h. The high photocatalytic behavior is rationalized using computations and spectroscopic studies that suggest the large internal electric field is responsible for efficient charge generation and anisotropic charge migration within the nanostructure, ultimately leading to efficient hydrogen evolution. The results are compelling and the work is certainly thorough. However, there are several shortcomings that should be addressed throughout the work before it can be published in Nature Communications. As my expertise is in optical spectroscopy I will focus my concerns to that topic.

General comments:

-Recent work by hydrogen evolution in similar perylenediimide-based nanoribbons by Stupp and co-workers is not mentioned or cited.

-Throughout the manuscript error bars/uncertainties on reported parameters are not reported. This makes reproducibility of the catalytic performance difficult to estimate.

Regarding the transient absorption spectroscopy:

-The explanation of the long-lived excited state in PTA vs PTCDA and how it relates to the generation of charge carriers is insufficient. The photocatalytic behavior clearly shows that long-lived charge is being generated, to be sure. But if the claim is that increased IEF creates a longer-lived excited state that then leads to an increased number of photogenerated charges (page 12), only monitoring the excited state (via ESA or PL) does not demonstrate this. Longer-lived excited states may simply result in higher fluorescence quantum yields and not result in higher free carriers. The authors can support their claim by monitoring instead features associated with the charge carriers, or demonstrate a contribution to this state at 900 nm from chemical redox or spectroelectrochemical measurements. These data would also elucidate the apparent spectral differences at early and long time delays in PTA between 450-600 nm in the bleach region (showing depopulation of the ground state) and would be much stronger evidence for long-lived charge separation in the TAS data.

-Depending on whether there is any contribution from the free carriers at 900 nm (which is unlikely in the case of perylene), the abbreviation of the excited-state lifetime in the sheets compared to the intrinsic monomeric lifetime from the time-resolved PL (see below) can be used to estimate the yield of carrier production. How does the efficiency compare to the measured/estimated efficiencies from the other techniques? What is limiting this efficiency?

-The authors discuss anisotropic charge (i.e. CT exciton or free carrier) migration, however it is not clear to me that this is not due, at least in part, to anisotropic Frenkel exciton (effectively, excited-state) migration prior to dissociation. Can the authors comment on this? The above points may give some insight into this distinction.

-The plotting of the TAS kinetics is misleading in several ways. 1) In Figure 4d it is difficult to tell what is happening at early times. Perhaps presenting these data with a logarithmic time axis will make the comparison at early times more facile and will also likely emphasize the difference in long-time behavior as well. 2) Similarly, the normalization of the data appears to be mapping the min/max of the data to [0,1]; this gives the impression that the PTA data trend to zero and possibly go negative at time delays beyond the temporal window of the experiment (this latter possibility is not permitted based on the steady-state spectra presented in Figure S32). Instead, the raw data in Figure S31 suggest the spectral intensity plateaus at a value near 20-25% of the maximum. It would be better if the data were normalized by dividing by the maximum. 3) If the data were normalized from [0,1] as suspected, then fitting the data to the biexponential decay is likely to give incorrect information on the excited-state dynamics. For example, if there is a long-lived plateau that persists beyond the experimental window, forcing the fit to go to zero by the end of the window will miss this contribution, and thus alter the interpretation. A fit of the raw data should be shown to prove the correct choice of fitting model. Indeed, it may strengthen the case for long-lived charge carriers, as elongated ESA lifetimes indicate (Frenkel) exciton migration which can precede charge separation, and even provide an estimate of the yield for excitons that escape immediate decay.

-The reported excitation fluence is high, and with the high concentration of chromophores within the excitation volume significant nonlinearity in the kinetics due to exciton-exciton annihilation is expected. This nonlinearity will lead to non-exponential kinetics and will ultimately lower the singlet population available to perform catalysis, making comparison to the operational conditions of the devices difficult. This will impact the kinetic efficiency discussed above.

-Perhaps I missed it, but were the fluorescence quantum yields of these materials measured? The fluorescence quantum yields of the sheets speak directly to the efficiency of charge transfer.

-Incidentally, the number of reported significant figures on the excited-state lifetime (2058.99 ps, pg 11) is too high; such precision is meaningless without an accompanying uncertainty, especially when the lifetime approaches the maximum pump-probe delay in the experiment. Similarly, the precision for the PTCDA lifetime is also too high. See comment above about the choice of model.

-The comparison of the PL lifetimes between PTA and PTCDA does not report on the differences in recombination, as the authors claim (pg 13). In organic systems like these PL will almost exclusively originate from the local excited state (Frenkel exciton) of the chromophore, *not* the charge-separated state. Radiative recombination in organic systems is typically very weak. Hence, the PL lifetimes only report on the intrinsic excited-state lifetimes of the chromophores. It is also unclear if the time-resolved PL is on the nanosheets or solutions. The origin of this argument is similar to the use of ESA as an indicator of long-lived charge in the TAS experiment, which is incomplete.

There are a number of minor errors/typos that should be resolved:

-Figure S32: The long tail on the red edge of the absorption spectrum suggests a large amount of scatter. This should be corrected.

-Figures S31-32: "adsorption" should be "absorption"

-Figure S36: the number of reported significant figures in the PL lifetimes is much too high, with no consideration for the finite instrument response.

-Supplementary Material: page 45: the units on temperature are K, not K-1

Reviewer #2 (Remarks to the Author):

Guo et al. studied perylenetetracarboxylic acid nanosheet for photocatalytic hydrogen evolution in this work. The results shown in this work are interesting. But there are many experimental results and details needed to be further clarified.

1. The authors claimed the performance of PTA photocatalyst is "one of the most excellent properties of organic photocatalysts reported...". However, very high light intensity is used in this work, 530 mW.cm⁻² (450 mW.cm⁻² without UV), which is almost 4-5 times higher than the light normally be used in photocatalysis (100 mW.cm⁻² in full sun spectrum). To make the photocatalytic data is comparable (for ex. Supplementary table 3). I suggest the authors should remeasure all photocatalytic experiments by use of 100 mW.cm⁻² light intensity.

2. External quantum yield should be provided instead of or together with AQE

3. The details of experiments should be well provided. For example, no details on samples for impedance measurement and TAS test, no experimental details on IEF, etc.

4. Come to impedance test, it is unclear what the authors want to deliver. How can the photochemical data be related to electrochemical data? More discussion on this would be helpful.

5. Come to TAS test, it is very confusing what the authors compared with. Why choose 900 nm absorption? What does it stand for? Where are the full spectra of TAS test?

6. Related to point 4, the author used the kinetics of TAS at 900 to explain IEF data. How can the author be sure that the difference between 900 nm lifetime of PTA and PTCDA is not from morphology variation? Triplet state vs, singlet state? More discussion on this part would be appreciated.

7. When the authors discussed the effect on anisotropic separation of photogenerated charges, it was confusing me for a while, but I eventually understood (hopefully right) that the authors tried to use the hole to deposit Mn₂O₃ for the study. It would be better that the authors gave a clear motivation at the beginning of that paragraph.

8. The authors also claimed the production of hydrogen in a large scale. How about the stability? The authors only tested the system for 2.5 h, how can this data be reliable to calculate hydrogen production for 1 day? So, it would be good to have long-term stability test of the system

Tiny things:

Fig1 caption, there is a typo in "electron diffraction pattern"

NaIO₃ should be electron scavenger (it means taking electron) or not electron sacrificial agent (it means giving electron)

Reviewer #3 (Remarks to the Author):

The authors synthesized monolayer PTA nanosheet by facile hydrolysis-reassembly of PTCDA. A hydrogen evolution rate of 118.9 mmol g⁻¹ h⁻¹ was realized under full spectrum irradiation. The paper systematically investigated structure of PTA nanosheet catalyst, photocatalytic performance under various conditions, energy level, charge separation, anisotropic migration and hydrophily. In my point of view, it is a beautiful work, especially for the outstanding hydrogen evolution rate. Though such a system is a good reference for the further development of organic based photocatalyst, a few key parameters are still missing, and the meaning of this work needs to be further illustrated. It is not suitable for Nature Communications with the current version and major revision is suggested. Some other comments are listed as follow:

1. The authors argue that PTA nanosheet show outstanding HER because up-shifted CB position. In fact, it only solves half the problem because it also leads to raised VB, which is quite harmful to oxidation ability. So, what is the meaning to construct such a system, or is there any solution for this problem?
2. In common organic system, D-A structure is widely used for separation of excitons. In lack of such structure, this system seems to work like inorganic semiconductor. Is it possible to give a calculation of exciton binding energy and illustration about how excitons are separated in this system?
3. During the preparation of PTA, solution A was poured into solution B with rapid stirring to yield nanosheet in precipitate. Is the structure of nanosheet affected by stringing rate, ratio of the two solution or so on?
4. In the section "Charge separation and anisotropic migration", the author investigated IEF, excited state lifetime, carrier lifetime and carrier migration, but electron mobility, which determines how much electrons can move from the center to the edge of nanosheet, is missing. Please complete this data by either TFT, TOF, SCLC or other technique.

5. Figure 2d shows that PTA nanosheets undergo 13 hours reactions with negligible degeneration. It is known to all that oxygen-related species such as $\bullet\text{OH}$ and $\text{O}_2\bullet^-$ will significantly lead to destruction of organic compounds. Can the authors give an explanation for the stability of PTA nanosheets?
6. In general, acidic condition is in favor of hydrogen evolution because it reduces hydrogen evolution potential. However, with such a high CB position, I wonder if it is able to work in neutral and even basic condition.
7. In Supplementary Table 3, line Phosphonate-Based MOFs and line L-PyBT, light source and sacrificial are written wrongly.
8. There are several mistakes, line 171 and figure 3b for example, about hydrogen evolution potential. It is always 0 V versus RHE, or -0.413 V versus SHE when $\text{pH} = 7$.
9. In Figure 6b, there is arrow that indicates $-\text{COOH}$ was convert into $-\text{COOD}$. It is confusing that why the peak of $-\text{COOD}$ points upward. It is possible to give a detailed instruction?

Detailed Responses to Reviewers

Manuscript number: NCOMMS-21-38971A-Z

Manuscript Type: Article

Title: Perylenetetracarboxylic Acid Nanosheet with Robust Internal Electric Field and Anisotropic Charge Migration for Superior Photocatalytic Hydrogen Evolution

Correspondence Author: Yongfa Zhu

[The reviewer comments are shown in *italic*; Responses are in **blue**; all revisions in the Manuscript and Supplementary are in black]

Reviewer #1 (Remarks to the Author):

General Comment. The manuscript by Guo et al. details the synthesis of perylene tetracarboxylic acid (PTA) aggregates and their structural, photophysical, and photocatalytic behavior. The authors find that the the PTA forms nanosheet structures with nanolayer thickness. Upon irradiation, hydrogen evolution is seen at a high rate of 119 mmol/g/h. The high photocatalytic behavior is rationalized using computations and spectroscopic studies that suggest the large internal electric field is responsible for efficient charge generation and anisotropic charge migration within the nanostructure, ultimately leading to efficient hydrogen evolution. The results are compelling and the work is certainly thorough. However, there are several shortcomings that should be addressed throughout the work before it can be published in Nature Communications. As my expertise is in optical spectroscopy I will focus my concerns to that topic.

Response: We are very grateful to the Reviewer for the positive comments and the recommendation for publication of our work in Nature Communications. We also appreciate the Reviewer's valuable and professional suggestions that have helped us to improve our paper. We have taken all comments and suggestions into account. The Response to each question is described below.

Comment 1. Recent work by hydrogen evolution in similar perylenediimide-based nanoribbons by Stupp and co-workers is not mentioned or cited.

Response 1: Thank you for your introduction to these wonderful research work by Stupp and co-workers. According to your suggestion, we properly cite these articles in introduction as:

The Stupp's group has carried out groundbreaking research on the photocatalytic hydrogen production from perylenediimide based self-assembled supramolecular materials⁹⁻¹¹.

9. Weingarten, A. S. *et al.* Self-assembling hydrogel scaffolds for photocatalytic

hydrogen production. *Nature Chemistry* **6**, 964-970, (2014).

10. Dannenhoffer, A. J. *et al.* Growth of Extra-Large Chromophore Supramolecular Polymers for Enhanced Hydrogen Production. *Nano Letters* **21**, 3745-3752, (2021).

11. Weingarten, A. S. *et al.* Chromophore Dipole Directs Morphology and Photocatalytic Hydrogen Generation. *J Am Chem Soc* **140**, 4965-4968, (2018).

Comment 2. Throughout the manuscript error bars/uncertainties on reported parameters are not reported. This makes reproducibility of the catalytic performance difficult to estimate.

Response 2: Thank you for your suggestion, we have added error bars to the photocatalytic hydrogen production data based on our test results.

Fig. 2. Photocatalytic hydrogen evolution. (a) Photocatalytic HER under full spectrum ($\lambda \geq 300 \text{ nm}$, light intensity: $\sim 530 \text{ mW cm}^{-2}$), visible light ($\lambda \geq 420 \text{ nm}$, light intensity: $\sim 450 \text{ mW cm}^{-2}$) and AM1.5G, (100 mW cm^{-2}) of PTA. (b) AM1.5G (100 mW cm^{-2}) HER performance of PTA (2 mg) loaded on

nonwoven fabrics, inset: Optical image of PTA loaded on nonwoven devices. (c) The wavelength-dependent AQE for photocatalytic HER over PTA. (d) The stable photocatalytic HER performance of PTA under full spectrum. Reaction conditions: 7.0 mg photocatalysts, 4.6 wt% Pt as co-catalysts, 100 mL, 0.2 M ascorbic acid solution and the pH is 2.45.

Comment 3. The explanation of the long-lived excited state in PTA vs PTCDA and how it relates to the generation of charge carriers is insufficient. The photocatalytic behavior clearly shows that long-lived charge is being generated, to be sure. But if the claim is that increased IEF creates a longer-lived excited state that then leads to an increased number of photogenerated charges (page 12), only monitoring the excited state (via ESA or PL) does not demonstrate this. Longer-lived excited states may simply result in higher fluorescence quantum yields and not result in higher free carriers. The authors can support their claim by monitoring instead features associated with the charge carriers, or demonstrate a contribution to this state at 900 nm from chemical redox or spectroelectrochemical measurements. These data would also elucidate the apparent spectral differences at early and long time delays in PTA between 450-600 nm in the bleach region (showing depopulation of the ground state) and would be much stronger evidence for long-lived charge separation in the TAS data.

Response 3: We are grateful to the reviewers for highly valuable advice on TAS. We have added monitoring of the characteristics associated with the charge carriers by TAS to support long-lived charge generation. We attribute ESA peaks by adding photogenerated electron and hole scavengers. And ultrafast broadband transient absorption (TA) is measured using mid-infrared (MIR) probe pulses to observe photogenerated electron dynamics. Besides, in order to exclude solvent interference with the ground state, we carried out TAS of PTA and PTCDA in form of solid coated films. The manuscript has been revised as follows:

Description in Supplementary: First, the ESA signals representing photogenerated holes and photogenerated electrons were assigned in the TAS in the presence of hole and electron scavengers ⁶, respectively. The ESA signal at ~650 nm changes significantly, with a fitted decay lifetime of 252.8 ps in the absence of any scavenger (Supplementary Figure 33a.), accelerating to 160.6 ps in the presence of the hole scavenger ascorbic acid (Supplementary Figure 33c.), and slowing to 338.4 ps in the presence of the electron scavenger AgNO₃ (Supplementary Figure 33b). This validates that the ESA signal at ~650 nm can be used to analyze the decay kinetics of photogenerated holes.

Supplementary Figure 33. TAS of (a) PTA (b) PTA in AgNO₃ (c) PTA in ascorbic acid at different time delays after nanosecond laser excitation at 400 nm (in the form of coating film). (d) The transient absorption decay kinetics at ~650 nm of PTA and PTA in photogenerated holes and electron scavengers.

Probing the decay kinetics of photogenerated electrons in the mid-infrared region (MIR). In the absence of any scavenger, the decay time of the signal at ~5000 nm was

3.3 ps, which was accelerated to 2.3 ps by the presence of AgNO₃ electron scavenger and further slowed to 5.3 ps by the presence of ascorbic acid hole scavenger (Supplementary Figure 34).

Supplementary Figure 34. The transient absorption decay kinetics at ~5000 nm for (a) PTA and PTCDA, (b) PTA and PTA in photogenerated holes and electron scavengers (in the form of coating film). 0.2 M ascorbic acid and AgNO₃ act as scavengers for holes and electrons respectively.

Description in manuscript: As a result of IEF driving effect, the photogenerated carriers are conveniently separated. We probed the charge kinetics by femtosecond-resolved transient absorption spectroscopy (TAS)^{68,69}. The positive absorbance represents the excited state absorbs (ESA) of S1-Sn (Supplementary Fig. 32)^{70,71}. The ESA signals representing photogenerated holes (at ~650 nm) and electrons (at ~5000nm) were assigned in the presence of holes and electron scavengers, respectively (Supplementary Fig. 33-34). The measured changes in the (Mid-infrared) MIR region are mainly caused by the absorption of photogenerated carriers^{72,73} and further ruled out the possibility of exciton assignment. We have fitted the decay kinetics of photogenerated charge using a second-order exponential fit, where τ_1 indicates that the photogenerated carries are rapidly compounded after exciton dissociation and τ_2 indicates that the photogenerated carries are trapped during migration after dissociation (Supplementary Notes 6). Compared to PTCDA, the

recombination of photogenerated carriers is inhibited by the IEF (presented as the prolongation of τ_1). And the τ_2 lifetimes of photogenerated holes (at ~ 650 nm) and photogenerated electrons (at ~ 5000 nm) of PTA are prolonged by times 3.2 (Supplementary Figure 35) and 4.7 (Fig.4d), respectively. After excluding the effect of morphology on the behaviour of photogenerated charges (Supplementary Notes 5), this is interpreted as the longer decay results from the IEF driving force increase the number of surviving photogenerated charge.

Supplementary Figure 35. The transient absorption decay kinetics at 650 nm of PTA and PTCDA (in the form of coating film). The photogenerated hole is represented at 650 nm and the delay is slowed down for the PTA compared to the PTCDA.

Fig. 4. IEF drives charge separation. (a) Excited charge density difference of PTA and PTCDA cell (Set with carbonyl oxygen). (b) The calculated intensity of internal electric field, (c) the impedance and (d) transient absorption decay kinetics at ~5000 nm of PTCDA and PTA.

References

6. Wolff, C. M. *et al.* All-in-one visible-light-driven water splitting by combining nanoparticulate and molecular co-catalysts on CdS nanorods. *Nature Energy* **3**, 862-869, (2018).
68. Wei, Z. *et al.* Steering Electron-Hole Migration Pathways Using Oxygen Vacancies in Tungsten Oxides to Enhance Their Photocatalytic Oxygen Evolution Performance. *Angewandte Chemie-International Edition* **60**, 8236-8242, (2021).
69. Liu, F. *et al.* Direct Z-Scheme Hetero-phase Junction of Black/Red Phosphorus for Photocatalytic Water Splitting. *Angew. Chem. Int. Ed.* **58**, 11791-11795, (2019).

70. Vdović, S. *et al.* Excited state dynamics of β -carotene studied by means of transient absorption spectroscopy and multivariate curve resolution alternating least-squares analysis. *Physical Chemistry Chemical Physics* **15**, 20026-20036, (2013).

71. Chen, X. J., Wang, J., Chai, Y. Q., Zhang, Z. J. & Zhu, Y. F. Efficient Photocatalytic Overall Water Splitting Induced by the Giant Internal Electric Field of a g-C₃N₄/rGO/PDIP Z-Scheme Heterojunction. *Adv. Mater.* **33**, 7, (2021).

Comment 4. Depending on whether there is any contribution from the free carriers at 900 nm (which is unlikely in the case of perylene), the abbreviation of the excited-state lifetime in the sheets compared to the intrinsic monomeric lifetime from the time-resolved PL (see below) can be used to estimate the yield of carrier production. How does the efficiency compare to the measured/estimated efficiencies from the other techniques? What is limiting this efficiency?

Response 4: Thank you for your comments. Based on your suggestion, we estimate the yield of the carrier production as shown in Supplementary Notes 1:

Supplementary Notes 2: Estimate the yield of carrier production

We estimate the yield of the carrier production by comparing the photogenerated electron lifetime with the stimulated emission (SE) delay from TAS data. Under equivalent pump light excitation conditions, the photogenerated electron delay of the PTA nanosheets is 134.2 ps at ~5000 nm and the SE delay at ~556 nm is 1421.9 ps, the ratio of which is roughly estimated to be 9.4% for carrier production.

$$Q = \frac{\text{Delay of photogenerated electron}}{\text{Delay of stimulated emission}} = \frac{134.2}{1421.9} = 9.4\%$$

Supplementary Figure 45. (a) TAS of PTA at different time delays after nanosecond laser excitation at 400 nm (1×10^{-5} M, aqueous solution). (b) transient absorption decay kinetics at 5000 nm and 556 nm of PTA.

As shown in Figure 2c in manuscript, the apparent quantum efficiency of hydrogen production of the PTA nanosheets at 400 nm excitation was 4.8%, which is lower than the estimated carriers generation rate (9.4%). The carriers produced were not fully used for hydrogen production, which may be attributed to processes such as carriers are used to generate reactive oxygen species. As shown in the electron paramagnetic resonance (EPR) measurement (Supplementary Fig.46), where the photogenerated electrons of PTA are trapped by dissolved oxygen to form $\cdot\text{O}_2^-$ in an air atmosphere. It further reacts with H^+ to form $\cdot\text{OH}$.

Supplementary Figure 46. EPR test of the hydroxyl radical ($\cdot\text{OH}$) and superoxide radical ($\cdot\text{O}_2^-$) signals captured by DMPO (0.1 M) in air atmosphere. It shows

that $\cdot\text{OH}$ and $\cdot\text{O}_2^-$ are produced by PTA nanosheets after light exposure in an air atmosphere.

Comment 5. The authors discuss anisotropic charge (i.e. CT exciton or free carrier) migration, however it is not clear to me that this is not due, at least in part, to anisotropic Frenkel exciton (effectively, excited-state) migration prior to dissociation. Can the authors comment on this? The above points may give some insight into this distinction.

Response 5: We thank the Reviewer for pointing out this issue. The present work monitors the production sites of photogenerated electrons and holes on PTA nanosheets by the directional deposition of Pt and Mn₂O₃, thus predicting the directional migration of photogenerated charges. However, for electrically neutral excitons, the internal electric field is not sufficient to drive their directional migration⁷⁷, i.e. the directional deposition of Pt and Mn₂O₃ is not contributed by Frenkel exciton migration. We add description in Manuscript as:

Obviously, the excitons do not move in whole under the electric field action. Therefore, the directional deposition of Pt and Mn₂O₃ is induced by carriers in the dissociated state. The carrier production rate was estimated to be 9.4% (Supplementary Notes 2), and the detection of carriers by MIR-TAS confirms this view.

Reference 77. Jin, S. *et al.* Surface modification boosts exciton extraction in confined layered structure for selective oxidation reaction. *Science China Chemistry* **64**, 1964-1969, (2021).

Comment 6. The plotting of the TAS kinetics is misleading in several ways. 1) In

Figure 4d it is difficult to tell what is happening at early times. Perhaps presenting these data with a logarithmic time axis will make the comparison at early times more facile and will also likely emphasize the difference in long-time behavior as well. 2) Similarly, the normalization of the data appears to be mapping the min/max of the data to [0,1]; this gives the impression that the PTA data trend to zero and possibly go negative at time delays beyond the temporal window of the experiment (this latter possibility is not permitted based on the steady-state spectra presented in Figure S32). Instead, the raw data in Figure S31 suggest the spectral intensity plateaus at a value near 20-25% of the maximum. It would be better if the data were normalized by dividing by the maximum. 3) If the data were normalized from [0,1] as suspected, then fitting the data to the biexponential decay is likely to give incorrect information on the excited-state dynamics. For example, if there is a long-lived plateau that persists beyond the experimental window, forcing the fit to go to zero by the end of the window will miss this contribution, and thus alter the interpretation. A fit of the raw data should be shown to prove the correct choice of fitting model. Indeed, it may strengthen the case for long-lived charge carriers, as elongated ESA lifetimes indicate (Frenkel) exciton migration which can precede charge separation, and even provide an estimate of the yield for excitons that escape immediate decay.

Response 6: We apologize for misrepresenting the plotting of the TAS kinetics, and thank the reviewer's professional suggestion. We have therefore optimized the plotting of the TAS kinetics and carefully considered the choice of model. Following your suggestion, we have used a logarithmic time axis for the plotting of the TAS kinetics, and by dividing by the maximum value to normalize, such as Figure 4d and Supplementary Figure 33-35. Besides, we address the correctness of the fitted model as shown in Supplementary Notes 6.

Supplementary Figure 33. TAS of (a) PTA (b) PTA in AgNO_3 (c) PTA in ascorbic acid at different time delays after nanosecond laser excitation at 400 nm (in the form of coating film). (d) The transient absorption decay kinetics at ~ 650 nm of PTA and PTA in photogenerated holes and electron scavengers.

Supplementary Figure 34. The transient absorption decay kinetics at ~ 5000 nm for (a) PTA and PTCDA, (b) PTA and PTA in photogenerated holes and electron scavengers (in the form of coating

film). 0.2 M ascorbic acid and AgNO_3 act as scavengers for holes and electrons respectively.

Supplementary Figure 35. The transient absorption decay kinetics at 650 nm of PTA and PTCDA (in the form of coating film). The photogenerated hole is represented at 650 nm and the delay is slowed down for the PTA compared to the PTCDA.

Fig. 4. IEF drives charge separation. (a) Excited charge density difference of PTA and PTCDA cell (Set

with carbonyl oxygen). (b) The calculated intensity of internal electric field, (c) the impedance and (d) transient absorption decay kinetics at ~5000 nm of PTCDA and PTA.

Supplementary Notes 6: Rationalization of second-order exponential functions to fit TAS kinetic processes

We have fitted the decay kinetics of photogenerated charge using a second-order exponential fit, where τ_1 indicates that the photogenerated carries are rapidly compounded after exciton dissociation and τ_2 indicates that the photogenerated carries are trapped during migration after dissociation. We fit the TAS data according to the second-order exponential and all the R^2 can reach above 0.95, and the residual analysis shows that the mathematical model has a high confidence level.

Supplementary Figure 52. Residual scores from second-order exponential fits to TAS data for PTA and PTCDA.

Comment 7. The reported excitation fluence is high, and with the high concentration of chromophores within the excitation volume significant nonlinearity in the kinetics due to exciton-exciton annihilation is expected. This nonlinearity will lead to non-exponential kinetics and will ultimately lower the singlet population available to perform catalysis, making comparison to the operational conditions of the devices difficult. This will impact the kinetic efficiency

discussed above.

Response 7: We agree with the reviewers that exciton-exciton annihilation does occur in organic molecular systems. However, for the PTA nanosheets with an ordered crystal structure in this paper, the effect of exciton-exciton annihilation for second-order exponential fit can be ignored (As shown in Response 6, R2 is above 0.95). We give details in Supplementary:

Supplementary Notes 6: Rationalization of second-order exponential functions to fit TAS kinetic processes

We have fitted the decay kinetics of photogenerated charge using a second-order exponential fit, where τ_1 indicates that the photogenerated carries are rapidly compounded after exciton dissociation and τ_2 indicates that the photogenerated carries are trapped during migration after dissociation.

In this study, the PTA nanosheets show an ordered crystal structure, which was confirmed by XRD as well as by electron diffraction patterns (Figure 1). According to the previous reports¹², excitons of highly crystalline organic materials are not prone to exciton-exciton annihilation because of the large effective volume, which is expected to give favourable exponential dynamics. From the results, we fit the TAS data according to the second-order exponential and all the R^2 can reach above 0.95, and the residual analysis shows that the mathematical model has a high confidence level.

Supplementary Figure 52. Residual scores from second-order exponential fits to TAS data for PTA and

PTCDA.

Reference 2. Evans, A. M. *et al.* Seeded growth of single-crystal two-dimensional covalent organic frameworks. *Science* **361**, 52-57, (2018).

Comment 8. Perhaps I missed it, but were the fluorescence quantum yields of these materials measured? The fluorescence quantum yields of the sheets speak directly to the efficiency of charge transfer.

Response 8: Thanks for your suggestion that we have measured the fluorescence quantum yields. We give measurement details in Supplementary and revised in Manuscript.

Description in Manuscript:

And the π - π channel enhances the excitonic coupling of the system, allowing the PTA nanosheets to exhibit a fluorescence quantum yield of 4.9% in water (Supplementary Notes 1)⁹.

Reference 9. Weingarten, A. S. *et al.* Self-assembling hydrogel scaffolds for photocatalytic hydrogen production. *Nature Chemistry* **6**, 964-970, (2014).

Details in Supplementary:

Supplementary Notes 1: Fluorescence quantum yields calculations

Relative quantum yield calculations were performed using a modified procedure by Williams et al⁷. The quantum yield of a sample relative to a reference compound can be calculated using the following equation:

$$Q_S = Q_R \left(\frac{A_R}{A_S} \right) \left(\frac{E_S}{E_R} \right) \left(\frac{I_R}{I_S} \right) \left(\frac{n_S^2}{n_R^2} \right) \quad (1)$$

Q_S - Quantum yield of the sample

Q_R - the known quantum yield of the reference

A_R - Absorption of the reference at a single excitation wavelength

A_S - absorption of the sample at a single excitation wavelength

E_R - the combined emission of the reference in its emission range

E_S - the combined emission of the sample over its emission range

I_R - the intensity of the excitation wavelength of the reference

I_S - the intensity of the excitation wavelength of the sample

n_R - refractive index of the solvent of the reference

n_S - refractive index of the solvent of the sample

If the excitation wavelengths of the sample and the reference are the same, the equation simplifies to:

$$Q_S = Q_R \left(\frac{A_R}{A_S} \right) \left(\frac{E_S}{E_R} \right) \left(\frac{n_S^2}{n_R^2} \right) \quad (2)$$

This calculation can be done using a single concentration, or a range of concentrations.

This equation uses the slope of the integrated emission intensity versus absorbance curve at the excitation wavelength, avoiding errors in individual measurements.

$$Q_S = Q_R \left(\frac{\text{Slope}_S}{\text{Slope}_R} \right) \left(\frac{n_S^2}{n_R^2} \right) \quad (3)$$

We used the reported $QY = 0.94$ for rhodamine 6G in ethanol as a control compound. Typically, fluorescence and absorbance measurements were made by injecting 3.5 mL of the solution into a 10 mm quartz cuvette. Absorbance at 488 nm was tested. The emission spectra were collected by excitation at 488 nm and the fluorescence peaks in the 500-700 nm range were integrated.

To minimize the effect of self-absorption effects, a lower concentration of the substance to be measured was used. An ethanolic solution of 0.02-0.1 mmol/L of rhodamine 6G ($n = 1.361$) was used as a reference. The 6.7-33.4 $\mu\text{mol/L}$ of PTA water ($n = 1.333$) dispersion was used as the sample to be tested. The quantum yields for PTA in water were 4.9%.

Supplementary Figure 44: Fluorescence integrated area versus absorbance curves for (a) reference rhodamine 6G in ethanol and (b) PTA in water. The solutions were excited at 488 nm and the spectra were collected between 500-700 nm. The estimated slopes used for quantum yield calculations are shown on each graph. The quantum yields for PTA in water were 4.9%.

Reference 7. Williams, A. T. R., Winfield, S. A. & Miller, J. N. Relative fluorescence quantum yields using a computer-controlled luminescence spectrometer. *Analyst* **108**, 1067-1071, (1983).

Comment 9. Incidentally, the number of reported significant figures on the excited-state lifetime (2058.99 ps, pg 11) is too high; such precision is meaningless without an accompanying uncertainty, especially when the lifetime approaches the maximum pump-probe delay in the experiment. Similarly, the precision for the PTCDA lifetime is also too high. See comment above about the choice of model.

Response 9: Thank you for the reminder that we have changed the accuracy of the lifetime to one decimal place (Unit: ps), considering that the accuracy of the TAS instrument is 0.037 ps.

Comment 10. The comparison of the PL lifetimes between PTA and PTCDA does not report on the differences in recombination, as the authors claim (pg 13). In organic systems like these PL will almost exclusively originate from the local excited state (Frenkel exciton) of the chromophore, *not* the charge-separated state. Radiative recombination in organic systems is typically very weak. Hence, the PL lifetimes only report on the intrinsic excited-state lifetimes of the chromophores. It is also unclear if the time-resolved PL is on the nanosheets or solutions. The origin of this argument is similar to the use of ESA as an indicator of long-lived charge in the TAS experiment, which is incomplete.

Response 10: We fully agree with the reviewers that PL lifetimes are difficult to fully and visually reflect the differences in charge recombination in organic systems. We have therefore removed the time response of the PL and replaced it with a more detailed and deterministic characterization of the charge carrier dynamics using transient absorption spectroscopy (as described in Response 3).

Comment 11. There are a number of minor errors/typos that should be resolved: -Figure S32: The long tail on the red edge of the absorption spectrum suggests a large amount of scatter. This should be corrected.

Response 11: Thank you for your suggestion, the absorption scattering does affect the judgement of the spectrum. And we have retested to avoid scattering.

Supplementary Figure 36. UV-vis absorption and photoluminescence spectroscopy (The excitation wavelength is 400 nm) of PTA.

-Figures S31-32: "adsorption" should be "absorption"

We are sorry for the spelling mistake, we have corrected it (The original Figures 31 and 32 are changed to 32 and 36 respectively in this revision).

Supplementary Figure 32. Transient absorption spectra (TAS) of (a) PTCDA and (b) PTA at different time delays after nanosecond laser excitation at 400 nm (1×10^{-5} M, aqueous solution).

Supplementary Figure 36. UV-vis absorption and photoluminescence spectroscopy (The excitation wavelength is 400 nm) of PTA.

-Figure S36: the number of reported significant figures in the PL lifetimes is much too high, with no consideration for the finite instrument Response.

Thank you for your comments. According to your Comment 10, PL lifetimes in organic systems make it difficult to provide a complete and visual response to differences in charge recombination. Therefore, we have removed the transient PL data.

-Supplementary Material: page 45: the units on temperature are K, not K-1

Thanks for your suggestion, we have corrected it:

T—absolute temperature, K

Reviewer #2 (Remarks to the Author):

General Comment. Guo et al. studied perylenetetracarboxylic acid nanosheet for photocatalytic hydrogen evolution in this work. The results shown in this work are interesting. But there are many experimental results and details needed to be further clarified.

Response: We are very thankful for the Reviewer's positive evaluation of our work and valuable advices that improve the quality of our manuscript. We have carefully considered all the questions that were raised. The Response to each question is described below.

Comment 1. The authors claimed the performance of PTA photocatalyst is "one of the most excellent properties of organic photocatalysts reported...". However, very high light intensity is used in this work, 530 mW.cm-2 (450 mW.cm-2 without UV), which is almost 4-5 times higher than the light normally be used in photocatalysis (100 mW.cm-2 in full sun spectrum). To make the photocatalytic data is comparable (for ex. Supplementary table 3). I suggest the authors should remeasure all photocatalytic experiments by use of 100 mW.cm-2 light intensity.

Response 1: Thank the Reviewer for pointing out this important question, we have supplemented the hydrogen production experiments with 100 mW cm⁻². This includes the optimized hydrogen production performance at AM 1.5G, 100 mW cm⁻², hydrogen production from solid-loaded PTA onto a non-woven fabric and 24-hour cycling performance. At the same time, the data in Table S3 are also updated to add the parameter of light intensity. Accordingly, we have revised our manuscript as follows:

“At AM 1.5G simulated sunlight (100 mW cm⁻²), the hydrogen evolution from PTA nanosheets reached 41.8 mmol g⁻¹ h⁻¹ (292.6 umol h⁻¹).”

“The HER performance of the device reaches $8.0 \text{ mmol m}^{-2} \text{ h}^{-1}$ under AM 1.5G. Assuming 8 hours of irradiation per day, it can produce about 1.4 L of hydrogen per square meter per day (Fig. 2b).”

Fig. 2. Photocatalytic hydrogen evolution. (a) Photocatalytic HER under full spectrum ($\lambda \geq 300 \text{ nm}$, light intensity: $\sim 530 \text{ mW cm}^{-2}$), visible light ($\lambda \geq 420 \text{ nm}$, light intensity: $\sim 450 \text{ mW cm}^{-2}$) and AM 1.5G, (100 mW cm^{-2}) of PTA. (b) AM1.5G (100 mW cm^{-2}) HER performance of PTA (2 mg) loaded on nonwoven fabrics, inset: Optical image of PTA loaded on nonwoven devices.

“...as well as the continuous hydrogen production over a 24-hour period without a significant decrease under AM 1.5G (Supplementary Fig. 18), indicating the stability of the PTA nanosheets.”

Supplementary Figure 18. The stable photocatalytic HER performance of PTA under AM1.5G, (100 mW cm^{-2}). Reaction conditions: 7.0 mg photocatalysts, 4.6 wt% Pt as co-catalysts, 100 mL, 0.2 M ascorbic acid solution and the pH is 2.45.

Comment 2. External quantum yield should be provided instead of or together with AQE

Response 2: Thank you for your comments. The apparent quantum yield (AQY) is the ratio of the number of photogenerated carriers used for H₂ production to the number of photons incident on photocatalyst from the outside. It is also known as the apparent quantum efficiency (AQE), or sometimes as the external quantum efficiency (EQE)⁴.

On the other hand, the internal quantum yield (IQY) may be calculated from AQY if the number of absorbed photons by the particle photocatalyst is known. However, considering that photons are scattered by the photocatalyst particles, it is difficult to determine the internal quantum efficiency at this stage. Therefore, AQY is the most important parameter for assessing the efficiency of a photocatalyst.

We give a more detailed description for external quantum yield (AQY) in the Methods section of the manuscript (p. 20):

The apparent quantum efficiency (AQE), or sometimes as the external quantum efficiency (EQE), and can be described by Equation 2. The apparent quantum efficiency (AQE) for PTA photocatalytic hydrogen evolution: A 300W xenon lamp with a bandpass filter was used as the light source, and the irradiation area was controlled to be 1.0×1.0 cm². The AQE was calculated as follow equation:

$$\text{AQE (\%)} = \frac{\text{the number of evolved } H_2 \text{ molecules} \times 2}{\text{number of incident photons}} \times 100\% \quad (2)$$

Reference 4. Wang, Z. et al. Efficiency Accreditation and Testing Protocols for Particulate Photocatalysts toward Solar Fuel Production. *Joule* 5, 344-359, (2021).

Comment 3. The details of experiments should be well provided. For example, no details on samples for impedance measurement and TAS test, no experimental details on IEF, etc.

Response 3: Thank you for careful suggestion. We have added more experimental details for completeness, including sample details for impedance measurements and TAS tests, the surface photogenerated voltage and experimental details for IEF.

Samples for impedance measurement: Photoelectrochemical measurements were recorded on a CHI-660 E (China) electrochemical workstation with a standard three-electrode cell, including reference electrode (Ag/AgCl), counter electrode (Pt wire) and working electrodes (as-prepared samples covered ITO glass). Na₂SO₄ (0.1 mol L⁻¹) was used as electrolyte solution. For the working electrodes, 1 mg sample was dispersed in water to obtain a slurry. Then the slurry was coated onto the ITO glass and dried naturally (Supplementary Fig.43). Electrochemical impedance experiments: EIS spectra were obtained by perturbing the system with an AC signal of 1000 KHz to 10 KHz at an amplitude of 10 mV. The equivalent circuit method was used to resolve the charge transfer resistance of the catalyst.

Supplementary Figure 43. Cross-sectional SEM image of PTA photoelectrode.

Sample preparation method for TAS test: TAS for testing solid coating films: to

test the TAS of PTA, 1 mg of the sample is dispersed in 2 mL of deionized water and sonicated to disperse well. It is coated to the substrate in three times, and each coating needs to be dried before the next drop. To test the TAS of PTA with scavengers of photogenerated electrons and holes, 0.2 mol/L ascorbic acid or AgNO₃ was used as the scavenger to replace the above deionized water. For samples measured in the visible region, quartz glass was used as a substrate. For samples measured in the mid-infrared (MIR) region, CaF₂ plates were used as substrates.

Testing the TAS of PTA dispersed in water: 1 × 10⁻⁵ M of PTA was placed in water, dispersed well by sonication and placed in a cuvette to be measured. See Supplementary Fig. 32-35 for the TAS involved in this study.

Supplementary Methods 2.

Determination of internal electric field (IEF).

The IEF magnitude of samples was calculated by using the following equation developed by Kanata et al.¹³⁻¹⁷

$$F = (-2V_s\rho/\varepsilon\varepsilon_0)^{1/2} \quad (5)$$

Where F is the internal electric field magnitude, V_s is the surface potential, ρ is the surface charge density, ε is the low-frequency dielectric constant, and ε₀ is the vacuum dielectric constant. The above equation reveals that the IEF magnitude is mainly determined by the surface voltage and the charge density because assuming other parameters are constant for perylene-based materials.

The surface potential V_s is obtained by Kelvin Probe Force Microscopy (KPFM) in the mode of surface potential.

Supplementary Figure 59. Schematic diagram of Kelvin Probe Force Microscopy (KPFM) testing the surface potential of samples

Kelvin Probe Force Microscopy (KPFM) is a scanning technique for probing the electrostatic potential distribution on the surface of a sample. Specifically, a known bias is applied between the sample and the tip of the needle (Supplementary Fig.59), and the surface potential distribution of the sample is then measured by KPFM.

The surface charge density ρ is tested by the zeta potential and then calculated using the model given by Gouy-Chapman ¹⁸:

$$\sigma = \sqrt{8kT\varepsilon\varepsilon_0 n} \sinh\left(\frac{ze_0\psi_0}{2kT}\right) \quad (3)$$

Among them, for the smaller Zeta potential value ($|\psi_0| < 50mV$), there are approximately^{11,12}:

$$\psi_0 = \zeta(1 + D/a_1)e^{kD} \quad (4)$$

σ —surface charge density, C/m^2

k —the Boltzmann constant, $J \cdot K^{-1}$

T —absolute temperature, K

n —Number of electrolytes per unit volume, m^{-3}

e_0 —electron charge, C

z —Electrolyte valence

ψ_0 —Surface potential, **V**

a_1 —Particle Stokes radius, **m**

ζ —zeta potential, it was measured at a concentration of 0.1 g/L samples, a NaCl concentration of 10 mmol/L and a temperature of 25 ° C.

κ^{-1} —Debye length, **nm**

D—Distance from Sliding Layer to Particle Surface

Therefore, we can simplify the IEF intensity as a function of the surface potential V_s and Zeta potential ζ . With the increases of V_s and ζ values, the strength of IEF increases, as shown below.

$$F = (AV_s \sinh(\frac{ze_0\zeta(1 + \frac{D}{a_1})e^{\kappa D}}{2kT}))^{\frac{1}{2}} \quad (5)$$

Reference 13-18.

13 Li, J., Zhang, L., Li, Y. & Yu, Y. Synthesis and internal electric field dependent photoreactivity of Bi₃O₄Cl single-crystalline nanosheets with high {001} facet exposure percentages. *Nanoscale* **6**, 167-171, (2014).

14 Li, J., Zhan, G., Yu, Y. & Zhang, L. Superior visible light hydrogen evolution of Janus bilayer junctions via atomic-level charge flow steering. *Nat Commun* **7**, 11480, (2016).

15 Pierre Lefebvre, J. A. g., Bernard Gil, and Henry Mathieu. Time-resolved photoluminescence as a probe of internal electric fields in GaN-(GaAl)N quantum wells. *PHYSICAL REVIEW B* **59**, 5, (1999).

16 Jin Seo Im, H. K., J. Off, A. Sohmer, F. Scholz, and A. Hangleiter. Reduction of oscillator strength due to piezoelectric fields in GaN/Al_xGa_{1-x}N quantum wells. *PHYSICAL REVIEW B* **57**, 3, (1997).

17 Morello, G. *et al.* Intrinsic optical nonlinearity in colloidal seeded grown CdSe/CdS nanostructures: Photoinduced screening of the internal electric field. *Physical Review B* **78**, 8, (2008).

18 Shaw, D. J. *Introduction to colloid and surface chemistry*. 170-180 (London ; Butterworths, FMT:BK.3rd ed., 1980).

Supplementary Methods 3.

Surface photogenerated voltage (SPV) measurement mechanism.

The surface photovoltage (SPV) spectra were detected by a home-built apparatus, equipped with a lock-in amplifier (SR830, USA) synchronized with a xenon light source (CHF-XQ500W). The width of the slit is set to 3mm. The structure of the photovoltage sample cell is ITO glass - sample - ITO glass.

For N-type semiconductors, the Fermi energy level is higher than the surface state energy level. This leads to the migration of electrons from the native phase to the surface until the Fermi energy level reaches equilibrium, resulting in the accumulation of negative charge at the surface. Under light irradiation, photogenerated charge migrates to the surface. The SPV signal is the difference between the surface voltage in the light and the surface voltage in the dark. Therefore, the stronger the SPV signal, the better the separation of the photogenerated carriers.

Comment 4. Come to impedance test, it is unclear what the authors want to deliver. How can the photochemical data be related to electrochemical data? More discussion on this would be helpful.

Response 4: We thank the Reviewer for raising the important issue. The electrochemical impedance spectrum (EIS) is the variation of AC impedance with frequency for an electrochemical system under DC polarization. At the same frequency, the larger the diameter of the impedance circle, the smaller the corresponding capacitance value and the larger the impedance value of the resulting Faraday current. That is, the relative size of the radius of the circle corresponds to the magnitude of the charge transfer resistance of the working electrode, reflecting the migration efficiency of the photogenerated carriers of the photocatalyst. Thus, although the impedance is obtained by electrochemical testing, it reflects the charge transfer resistance of the working electrode itself,

which can be used to represent the intrinsic properties of the photocatalyst.

The equivalent circuit method is a commonly used method for processing impedance data. In this work, the charge transfer process of the working electrode is monitored using an equivalent circuit as shown in the inset of Figure 4c to simulate the electrochemical process of the working electrode. Where R_{Ω} , C_d , R_{ct} , Z_w is solution resistance, double layer capacitance at the research electrode interface, charge transfer resistance, Warburg impedance respectively. As demonstrated in Figure 4c, the radius of the circle is significantly reduced when PTA is used as the working electrode, and the equivalent circuit method fits a charge transfer resistance of 36 Ω and 21770 Ω for PTA and PTCDA respectively, indicating that the carrier migration resistance of PTA is lower.

Comment 5. Come to TAS test, it is very confusing what the authors compared with. Why choose 900 nm absorption? What does it stand for? Where are the full spectra of TAS test?

Response 5: We apology that we have not provided a clear description of the TAS test results. In this revision we have supplemented the TAS test with a clearer description in Manuscript and Supplementary:

Description in manuscript: As a result of IEF driving effect, the photogenerated carriers are conveniently separated. We probed the charge kinetics by femtosecond-resolved transient absorption spectroscopy (TAS)^{68,69}. The positive absorbance represents the excited state absorbs (ESA) of S1-Sn (Supplementary Fig. 32)^{70,71}. The ESA signals representing photogenerated holes (at ~650 nm) and electrons (at ~5000nm) were assigned in the presence of holes and electron scavengers, respectively (Supplementary Fig. 33-34). The measured changes in the (Mid-infrared) MIR region are mainly caused by the absorption of photogenerated carriers^{72,73} and further ruled out the possibility of exciton assignment. We have fitted

the decay kinetics of photogenerated charge using a second-order exponential fit, where τ_1 indicates that the photogenerated carries are rapidly compounded after exciton dissociation and τ_2 indicates that the photogenerated carries are trapped during migration after dissociation (Supplementary Notes 6). Compared to PTCDA, the recombination of photogenerated carriers is inhibited by the IEF (presented as the prolongation of τ_1). And the τ_2 lifetimes of photogenerated holes (at ~ 650 nm) and photogenerated electrons (at ~ 5000 nm) of PTA are prolonged by times 3.2 (Supplementary Figure 35) and 4.7 (Fig.4d), respectively. After excluding the effect of morphology on the behavior of photogenerated charges (Supplementary Notes 5), this is interpreted as the longer decay results from the IEF driving force increase the number of surviving photogenerated charge.

Fig. 4. IEF drives charge separation. (a) Excited charge density difference of PTA and PTCDA cell (Set with carbonyl oxygen). (b) The calculated intensity of internal electric field, (c) the impedance and (d) transient absorption decay kinetics at ~ 5000 nm of PTCDA and PTA.

Description in Supplementary:

First, the ESA signals representing photogenerated holes and photogenerated electrons were assigned in the TAS in the presence of hole and electron scavengers⁶, respectively. The ESA signal at ~650 nm changes significantly, with a fitted decay lifetime of 252.8 ps in the absence of any scavenger (Supplementary Figure 33a.), accelerating to 160.6 ps in the presence of the hole scavenger ascorbic acid (Supplementary Figure 33c.), and slowing to 338.4 ps in the presence of the electron scavenger AgNO₃ (Supplementary Figure 33b). This validates that the ESA signal at ~650 nm can be used to analyze the decay kinetics of photogenerated holes.

Supplementary Figure 33. TAS of (a) PTA (b) PTA in AgNO₃ (c) PTA in ascorbic acid at different time delays after nanosecond laser excitation at 400 nm (in the form of coating film). (d) The transient absorption decay kinetics at ~650 nm of PTA and PTA in photogenerated holes and electron scavengers.

Probing the decay kinetics of photogenerated electrons in the mid-infrared region (MIR). In the absence of any scavenger, the decay time of the signal at ~5000 nm was

3.3 ps, which was accelerated to 2.3 ps by the presence of AgNO_3 electron scavenger and further slowed to 5.3 ps by the presence of ascorbic acid hole scavenger (Supplementary Figure 34).

Supplementary Figure 34. The transient absorption decay kinetics at ~ 5000 nm for (a) PTA and PTCDA, (b) PTA and PTA in photogenerated holes and electron scavengers (in the form of coating film). 0.2 M ascorbic acid and AgNO_3 act as scavengers for holes and electrons respectively.

Supplementary Figure 35. The transient absorption decay kinetics at 650 nm of PTA and PTCDA (in the form of coating film). The photogenerated hole is represented at 650 nm and the delay is slowed down for the PTA compared to the PTCDA.

Comment 6. Related to point 4, the author used the kinetics of TAS at 900 to explain IEF data. How can the author be sure that the difference between 900 nm lifetime of PTA and PTCDA is not from morphology variation? Triplet state vs, singlet state? More discussion on this part would be appreciated.

Response 6: We thank the Reviewer for pointing out this important question. In this revision, we have retested and analyzed the TAS data. Unlike the original version, experiments with the addition of photogenerated electron and hole scavengers have confirmed that the absorbance at ~650 nm and ~5000 nm in TAS represent the lifetimes of photogenerated holes and electrons, respectively, as described in Response 5. This rule out the excitonic state, which eliminates concerns about the triplet and single-linear states. Delayed decays of photogenerated holes and photogenerated electrons serve as evidence for the role of the IEF. Regarding the influence of other factors on TAS dynamics, we believe the reviewer's suggestion that there is a potential influence of photocatalyst morphology on the motion of charge carriers is reasonable. We give details in the **Supplementary:**

Supplementary Notes 5: Excluding the influence of morphology on the behavior of photogenerated charges

To exclude the effect of morphology on photogenerated charge, we modulated the synthesis of PTA (Supplementary Fig. 50a-c) and PTCDA (Supplementary Fig. 51a-c) with different morphologies.

Supplementary Figure 50. Morphology of PTA regulated by (a) HCl, (b) H_3PO_4 and (c) CH_3COOH as acid. (d) Surface photovoltage and (e) current of PTA with different morphologies. Different morphologies were obtained by changing the type of acid (called solution B in the preparation method) used to prepare the PTA supramolecular. When HCl is used, the PTA formed is in strips of $\sim 2.0 \mu\text{m}$ in length. When H_3PO_4 was used, both short strips and sheets were available. The $\sim 520 \text{ nm}$ long PTA nanosheets shown in the main text were obtained using CH_3COOH .

The photogenerated charge properties are then explored for different morphologies of PTA. Surface photovoltage is a means of characterizing the accumulation of photogenerated charge on the surface. There are no significant differences in the SPV signals of the three morphologies of PTA (Supplementary Fig. 50d). In addition, their transient photocurrent signals are also relatively consistent (Supplementary Fig. 50e). This suggests that the effect of morphology on the photogenerated charge movement of PTA is minimal.

Supplementary Figure 51. SEM morphology of (a) commercial PTCDA, heat treatment at (b) 120°C and (c) 180°C. (d) Surface photovoltage and (e) current of PTCDA with different morphologies. The commercially available PTCDA was heated at 120°C and 180°C to obtain shapes with different lengths. The average rod lengths of commercially purchased, 120°C heat treated and 180°C heat treated PTCDA were 0.8 μm, 1.4 μm and 2.6 μm respectively.

To make the conclusions clearer, we also obtained PTCDA with different morphologies (Supplementary Fig. 51a-c) by heat treatment and characterized them in the same way as PTA. The PTCDA with different morphologies showed similar SPV (Supplementary Fig. 51d) and photocurrent (Supplementary Fig. 51e) signals to each other. This indicates that the effect of morphology on the photogenerated charge of PTCDA can also be neglected. Therefore, based on the above discussion, we conclude that the effect of morphology on the behavior of photogenerated charges can be ignored.

As revised in Manuscript:

After excluding the effect of morphology on the behaviour of photogenerated

charges (Supplementary Notes 5), this is interpreted as the longer decay results from the IEF driving force increase the number of surviving photogenerated charge.

Comment 7. When the authors discussed the effect on anisotropic separation of photogenerated charges, it was confusing me for a while, but I eventually understood (hopefully right) that the authors tried to use the hole to deposit Mn₂O₃ for the study. It would be better that the authors gave a clear motivation at the beginning of that paragraph.

Response 7: Thank you for the reminder that we have added the motivation for the study at the beginning of that paragraph in Manuscript:

We next sought to determine the orientation of IEF and its effect on anisotropic separation of photogenerated charge by photo-deposition experiment. With NaIO₃ as the electron scavenger, Mn²⁺ can be oxidized to Mn₂O₃ oxide by photogenerated holes. Taking ascorbic acid as the holes scavenger, the metal ion (Pt⁴⁺) is reduced to Pt by photogenerated electrons. Thus, the deposition sites of Mn₂O₃ and Pt can be used to explore the photogenerated holes and electron generation sites of PTA nanosheets.

Comment 8. The authors also claimed the production of hydrogen in a large scale. How about the stability? The authors only tested the system for 2.5 h, how can this data be reliable to calculate hydrogen production for 1 day? So, it would be good to have long-term stability test of the system

Response 8: Thank you for your suggestion, we supplemented the PTA nanosheets with a 1 day 24-hour hydrogen production in AM1.5G simulated sunlight (100mW cm⁻²). As shown in the Figure , there is no significant decrease in hydrogen production over 24 hours, indicating the stability of the PTA

nanosheets. Described in the manuscript as:

Furthermore, the HER activity could be retained to exceed 5 individual cycles (Fig. 2d), as well as the continuous hydrogen production over a 24-hour period without a significant decrease (Supplementary Fig. 18), indicating the stability of the PTA nanosheets.

Supplementary Figure 18. 24-hour continuous photocatalytic hydrogen production performance for PTA nanosheet under AM 1.5G, 100mW cm⁻².

Comment 9: Tiny things:

Fig1 caption, there is a typo in “electron diffraction pattern”

NaIO₃ should be electron scavenger (it means taking electron) or not electron sacrificial agent (it means giving electron)

Response 9: Thank you for the correction. It has been corrected to “electron diffraction pattern”. We have changed the expression to “electron scavenger” for NaIO₃, and we have double-checked the entire text and made corrections to similar descriptions.

Reviewer #3 (Remarks to the Author):

General Comments. The authors synthesized monolayer PTA nanosheet by facile hydrolysis-reassembly of PTCDA. A hydrogen evolution rate of 118.9 mmol g⁻¹ h⁻¹ was realized under full spectrum irradiation. The paper systematically investigated structure of PTA nanosheet catalyst, photocatalytic performance under various conditions, energy level, charge separation, anisotropic migration and hydrophily. In my point of view, it is a beautiful work, especially for the outstanding hydrogen evolution rate. Though such a system is a good reference for the further development of organic based photocatalyst, a few key parameters are still missing, and the meaning of this work needs to be further illustrated. It is not suitable for Nature Communications with the current version and major revision is suggested. Some other comments are listed as follow:

Response: We are very appreciative of the Reviewer's recommendation for publication and positive evaluation of our work. We also thank the Reviewer for the constructive comments that help us to enrich our paper. We have considered all the points that were raised. The Response to each question is described below

Comment 1. The authors argue that PTA nanosheet show outstanding HER because up-shifted CB position. In fact, it only solves half the problem because it also leads to raised VB, which is quite harmful to oxidation ability. So, what is the meaning to construct such a system, or is there any solution for this problem?

Response 1: We thank the Reviewer for pointing out this important question. In photocatalytic HER, the energy of the photogenerated hole corresponds to the potential at the top of the valence band. Although the presence of photogenerated holes is detrimental to the hydrogen production (water reduction) process, ascorbic acid is added as a hole trapping agent in the actual reaction. Ascorbic acid is easy to oxidize and the position of the photocatalyst VB only needs to be

sufficient to oxidize ascorbic acid. Conversely, a lowered valence band position (i.e. enhanced oxidation capacity) would instead allow the hydrogen produced to be oxidized. Therefore, for photocatalytic hydrogen production, an elevated VB position is favorable for hydrogen production.

Comment 2. In common organic system, D-A structure is widely used for separation of excitons. In lack of such structure, this system seems to work like inorganic semiconductor. Is it possible to give a calculation of exciton binding energy and illustration about how excitons are separated in this system?

Response 2: We thank the Reviewer for raising the important issue. We give theoretical calculations of exciton binding energies and illustrate the process of exciton dissociation in PTA nanosheets.

Details in Supplementary:

Supplementary Notes 3: The exciton binding energy of PTA

The Coulomb attraction energy of electrons and holes is used to evaluate the exciton binding energy of PTA through the Coulomb equation as follows ^{8,9}:

$$E_C = \iint \frac{\rho^{\text{hole}}(r_1)\rho^{\text{ele}}(r_2)}{|r_1 - r_2|} dr_1 dr_2 \quad (5)$$

where ρ represents the densities of the natural transition orbitals (NTOs) of the hole ρ^{hole} and the electron ρ^{ele} , respectively. The NTOs are derived by the Cubgen tool included in the Gaussian 09 suite.

Based on the time-dependent density functional theory (TD-DFT), the main transitions of the first five singlet excited states were determined by the vibrator strength. Exciton binding energy analysis was performed on PTA with different π - π stacking layers (Supplementary Figure 47). As expected, with the increase in the number of layers, the charge transfer excitation process is subsequently enhanced and

the exciton binding energy decreases from 4.41 eV to 0.72 eV. At lower exciton binding energies, excitons dissociate more readily into photogenerated electrons and photogenerated holes.

Further, when the number of layers is 6, the electrons of the PTA stacker are mainly enriched on the terminal thick ring, while the holes are enriched on the carboxyl group of the intermediate PTA molecule (Supplementary Figure 48). This further validates the IEF-induced carrier behavior in the PTA molecule, i.e., holes migrate to the carboxyl group driven by the IEF, while electrons migrate to the edges of the nanosheets through channels formed by the π - π stacking.

Supplementary Figure 47. exciton binding energy of PTA in different layer stacking.

Supplementary Figure 48. Excited state electron-hole distribution of PTA stack (layer number 6).

(Green equivalence plane represents electron distribution; blue equivalence plane represents hole distribution)

Reference

8. Kraner, S., Scholz, R., Plasser, F., Koerner, C. & Leo, K. Exciton size and binding energy limitations in one-dimensional organic materials. *JOURNAL OF CHEMICAL PHYSICS* **143**, (2015).

9. Kraner, S., Prampolini, G. & Cuniberti, G. Exciton Binding Energy in Molecular Triads. *JOURNAL OF PHYSICAL CHEMISTRY C* **121**, 17088-17095, (2017).

Revised in the manuscript:

In highly ordered PTA nanosheets, IEF acts on exciton dissociation to form carriers for further migration (Supplementary Notes 3).

Comment 3. During the preparation of PTA, solution A was poured into solution B with rapid stirring to yield nanosheet in precipitate. Is the structure of nanosheet affected by stringing rate, ratio of the two solution or so on?

Response 3: Thank you for pointing out this important problem. Following your suggestion, we have explored the effect of B and A concentration ratios, and stirring time on the morphology of PTA nanosheets. Details and discussion are given in the Supplementary, and revised in Manuscript.

Details in Supplementary:

Supplementary Notes 7: Factors affecting the morphology of PTA nanosheets

We have explored the effect of solution B and A concentration ratios, and stirring time on the morphology of PTA nanosheets. As described in Methods, KOH was used as the B solution and acetic acid as the A solution. When B: A was increased from

14:1 to 136:1, the resulting precipitates were still nanosheets, but the length of the nanosheets decreased from $\sim 4.1 \mu\text{m}$ to $\sim 0.6 \mu\text{m}$. This was attributed to the ionic strength and polarity affecting the nucleation of the crystals and the growth kinetics of the nanosheets.

Supplementary Figure 53. The morphology of PTA prepared under condition c (KOH): c (Acetic Acid) =14:1 (a), 34:1 (b) and 136:1 (c).

The precipitates formed the moment A was added to B. As the stirring time was extended from 10 min to 24 h, the precipitates were all two-dimensional nanosheets with no significant change in their morphology. Attributed to the thermodynamic equilibrium reached, the nanosheets were stable thermodynamic morphologies.

Supplementary Figure 54. The morphology of PTA prepared by stirring for (a) 10 min,

(b) 0.5 h (c) 2 h (d) 24 h.

Description in Manuscript:

For the assemblies of perylenetetracarboxylic acid molecules, the nanosheet morphology is thermodynamically stable (Supplementary Notes 7).

Comment 4. In the section “Charge separation and anisotropic migration”, the author investigated IEF, excited state lifetime, carrier lifetime and carrier migration, but electron mobility, which determines how much electrons can move from the center to the edge of nanosheet, is missing. Please complete this data by either TFT, TOF, SCLC or other technique.

Response 4: We thank the Reviewer for pointing out this important question. We have completed the electron mobility of PTA and PTCDA by SCLC technique.

Supplementary Notes 4: Electron mobility testing by SCLC (space charge limited current)

To evaluate the charge mobility, the electron mobility of PTCDA and PTA was measured by SCLC method¹⁵. Its J-V curve is shown in Supplementary Figure 30. The mobility values were calculated by the Mott–Gurney law for the sandwich-type device:

$$\mu = \frac{8JL^2}{9\varepsilon_0 \varepsilon_r V^2} \quad (6)$$

Where J, ε_r , ε_0 and L represent the current density, relative permittivity, the vacuum permittivity and the distance between the two electrodes, respectively. It is assumed that $\varepsilon_r=3$, and ε_0 is the same for all crystals and a value of 8.85×10^{-14} has been used from previous reports¹⁶. The electron mobility was calculated to be 1.1 and $3.2 \text{ cm}^2 \text{ V}^{-1} \text{ s}^{-1}$ for PTCDA and PTA, respectively. For PTA nanosheets, the enhanced

internal electric field and the long-range ordered structure contribute to their electron mobility.

Supplementary Figure 54. The plot of the current density (J) versus applied voltage (V) measured in a ~200 um and 1.5cm² area thick samples at room temperature in air.

Reference

15 An, Z. S. *et al.* High electron mobility in room-temperature discotic liquid-crystalline perylene diimides. *ADVANCED MATERIALS* **17**, 2580, (2005).

16 Xue, D. A Simple Method for Determining Dielectric Constants of Materials with the Similar Crystal Structure. *Chem. Res. Chin. Univ.* **11**, 4, (2000).

We describe this in the section “Charge separation and anisotropic migration” of manuscript as:

This is further confirmed by the enhanced electron mobility of the space charge limited current (SCLC) tests (Supplementary Notes 4).”

Comment 5. Figure 2d shows that PTA nanosheets undergo 13 hours reactions with negligible degeneration. It is known to all that oxygen-related species such as •OH

and $O_2\cdot^-$ will significantly lead to destruction of organic compounds. Can the authors give an explanation for the stability of PTA nanosheets?

Response 5: We thank the Reviewer for raising the important issue. We discuss the stability of PTA nanosheets in terms of their chemical and crystalline structure, and photocatalytic HER environment.

Supplementary Notes 8: Explain the stability of PTA nanosheets.

We explain the stability of PTA nanosheets in four ways. Firstly, the perylenetetracarboxylic acid molecules are assembled into long range and orderly PTA nanosheets by π - π stacking. Effective overlap occurs between the molecular orbitals of the perylene rings, resulting in stable aggregates. The photogenerated charges generated by PTA under light irradiation leave the domain through the π -orbitals of the interacting molecules, which makes PTA less susceptible to damage.

Secondly, the reactive oxygen species produced by PTA nanosheets under light irradiation were characterized by EPR, as shown in the Supplementary Figure 55, where the photogenerated electrons of PTA are trapped by dissolved oxygen to form $\cdot O_2^-$ in an air environment. It further reacts with H^+ to form $\cdot OH$. However, the production of superoxide and hydroxyl radicals was barely detected in the nitrogen environment (Supplementary Figure 56), proving that the production of these reactive oxygen species is dependent on the oxygen environment. The photocatalytic hydrogen production from PTA nanosheets in this study was tested in a vacuum environment, so that the generation of reactive oxygen components was strongly inhibited.

Supplementary Figure 55. EPR test of $\cdot\text{OH}$ (a) and $\cdot\text{O}_2^-$ (b) production by PTA nanosheets after 3 minutes of light exposure under air atmosphere

Supplementary Figure 56. EPR test of $\cdot\text{OH}$ and $\cdot\text{O}_2^-$ production by PTA nanosheets after 3 minutes of light exposure under nitrogen atmosphere

Furthermore, even though traces of oxygen remaining in the hydrogen production environment led to the generation of small amounts of reactive oxygen species in the light. The perylenetetracarboxylic acid molecule possesses the large conjugated structure of perylene, which are difficult for the reactive oxygen species to oxidize and decompose.

Finally, the presence of ascorbic acid in the photocatalytic hydrogen production half-reaction is highly oxidizable and the reactive oxygen species generated on the surface of the PTA nanosheets will preferentially react with the thermodynamically easy ascorbic acid.

In conclusion, the long-range ordered structure of PTA, and the hydrogen production environment that is not conducive to the generation of reactive oxygen radicals ensure the stability of PTA nanosheets.

We described the stable of PTA nanosheets in manuscript as: The stable π - π stacking of PTA nanosheets ensures their stability (Supplementary Notes 8).

Comment 6. In general, acidic condition is in favor of hydrogen evolution because it reduces hydrogen evolution potential. However, with such a high CB position, I wonder if it is able to work in neutral and even basic condition.

Response 6: We greatly appreciate the Reviewer's valuable suggestions. Based on your suggestion, we performed a hydrogen production reaction under neutral conditions on PTA nanosheets. The pH of the photocatalytic hydrogen production reaction system was increased by adding sodium ascorbate instead of partial ascorbic acid. Under simulated sunlight irradiation at AM1.5G (100 mW/cm²), the hydrogen production at pH 6.2 and 7.0 was 1.2 and 0.4 mmol g⁻¹ h⁻¹, respectively, which was a significant decrease compared to the 41.8 mmol g⁻¹ h⁻¹ achieved when using ascorbic acid exclusively as a vacancy scavenger (pH = 2.5). One is attributed to the increased hydrogen production overpotential with increasing pH, as you described, even though the PTA nanosheets have a higher CB position, which makes photocatalytic HER less accessible. The second is since the PTA nanosheets are supramolecular formed by the assembly of perylenetetracarboxylic acid molecules. This molecule contains four carboxyl groups that gradually dissolve when the pH increases, which dictates that PTA nanosheets are not suitable for hydrogen production environments at larger pH. Despite this, due to the ease of modification of the perylene molecules, a wider pH range of hydrogen production can be expected through structural modifications.

Figure. The performance of photocatalytic hydrogen production at pH = 6.2 and 7.0 under AM 1.5G, 100mW cm⁻².

Comment 7. In Supplementary Table 3, line Phosphonate-Based MOFs and line L-PyBT, light source and sacrificial are written wrongly.

Response 7: Thank you for the heads up, we have corrected them.

Comment 8. There are several mistakes, line 171 and figure 3b for example, about hydrogen evolution potential. It is always 0 V versus RHE, or – 0.413 V versus SHE when pH = 7.

Response 8: Thanks to your correction, we have revised the relevant descriptions of Line 171, Figure 3b and Supplementary Figure 24 to (vs. SHE, pH=7) and have double-checked the manuscript correction.

Comment 9. In Figure 6b, there is arrow that indicates -COOH was convert into -COOD. It is confusing that why the peak of -COOD points upward. It is possible to give a detailed instruction?

Response 9: Thank you for your close examination. Our representation at this point is fuzzier than expected. The two small peaks near 2220 cm⁻¹ and 2060 cm⁻¹ represent -COOD, thus creating a bulge at 2111 cm⁻¹ between them. We have added indicator arrows in Figure 6b to make this clearer and more unambiguous.

Figure 6b. ATR-FTIR spectra of PTA before and after in situ adsorbing D₂O.

REVIEWER COMMENTS

Reviewer #1 (Remarks to the Author):

The authors have largely addressed my initial concerns with their rebuttal. However a few issues remain:

1) The yield analysis performed using the time-resolved data is incorrect and substantially underestimates the yield of electron transfer. If the yield were only 9% then one would expect the observed lifetime in the aggregate to be much closer to that in the monomer, however the aggregate decays an order of magnitude faster, suggesting rapid electron transfer (and thus a high yield). The ratio of the lifetimes also does not take into account the competing kinetic decay channels present for the excited state. The correct way to estimate this yield is to extract the charge transfer rate constant (I'll call it "k2" here) from the observed rate constant in the aggregate: $k_1+k_2 = 1/(134 \text{ ps})$, where k_1 is the rate of monomer decay ($k_1 = 1/1422 \text{ ps}$), using their values, and then computing the efficiency of CS using that value: the estimated yield of charge transfer is $k_2/(k_1+k_2) \sim 90\%$. The authors should revise their estimate and discuss the discrepancy between this value and the $\sim 5\%$ efficiency of H₂ generation. Large discrepancies are not uncommon, as surface area limitations and charge recombination pathways in the aggregates all lead to kinetic competition with catalysis, but the current discussion is insufficient and the numbers are not obtained in a rigorous way.

2) The addition of the mid-IR TAS data is quite useful and provides a better, more spectrally specific probe of the free carriers. Perhaps I missed them among the other data in the Supplemental Materials, but I did not see the actual mid-IR TA spectra anywhere in the manuscript or SM. The authors should provide the full spectrally resolved mid-IR TAS data similar to how they have provided the data for the visible and NIR ranges.

3) Similarly, there are no experimental details given for their mid-IR TAS measurements or apparatus beyond the sample preparation, which makes comparisons of the experimental conditions difficult. Full experimental details should be given in the Supplemental Materials.

Reviewer #2 (Remarks to the Author):

The authors have well addressed all comments. The paper can be accepted as is.

Reviewer #3 (Remarks to the Author):

In the revised manuscript, the authors tried the best to increase the quality by adding more experiments and giving more discussions according to the comments from the referees. I recommend acceptance of this manuscript for publication on Natural Communications.

The following points need to be further considered and clarified.

1. About the exciton migration in the nanosheet, the authors just give some references (40-42) to address the advantage of the ultrathin crystal. Is it possible to give some measurements about the exciton migration length? In nanocrystal, some time very long migration distance could be observed. Indeed, in recent reports on nonfullerene electron acceptors, the typical exciton migration distance is about 50nm. So, I am wondering the excitons might migrate along the pi-pi stacking direction, since the CT coupling is important in these nanosheets.

2. In the introduction section, when the perylene serial materials was discussed as catalyst for hydrogen evolution, it is better to give a proper description on the strategy to use low energy sunlight since it represents one of the important progresses in this field (doi.org/10.1002/anie.202001231).

3. In Figure 1e left, the thickness of one layer is show as 2 nm (from 0.75 to 2.75 nm); however, the description in page 6 line 175 is 1.5 nm observed by AFM. To my knowledge, the distance between O-O atoms in PTCDA is about 1nm. So, please check the molecular size in the optimized molecular structure, and give a more accurate presentation in Figure 1e.

4. In Figure 2a, the hydrogen evolution rates were obtained by using different light sources. It will be better to give the spectra of these light sources.

5. In Figure 2c, the AQE is not zero in the spectral region of >550nm. Was it in the error range? How about the AQE values in the near IR region?

Detailed Responses to Reviewers

Manuscript number: NCOMMS-21-38971B

Manuscript Type: Article

Title: Perylenetetracarboxylic Acid Nanosheet with Robust Internal Electric Field and Anisotropic Charge Migration for Superior Photocatalytic Hydrogen Evolution

Correspondence Author: Yongfa Zhu

[The reviewer comments are shown in *italic*; Responses are in **blue**; all revisions in the Manuscript and Supplementary are in black]

Reviewer #1 (Remarks to the Author):

General Comment. The authors have largely addressed my initial concerns with their rebuttal. However a few issues remain:

Response: We are very thankful for the Reviewer's positive evaluation of our work and valuable advices that improve the quality of our manuscript. We have carefully considered all the questions that were raised. The Response to each question is described below.

Comment 1. The yield analysis performed using the time-resolved data is incorrect and substantially underestimates the yield of electron transfer. If the yield were only 9% then one would expect the observed lifetime in the aggregate to be much closer to that in the monomer, however the aggregate decays an order of magnitude faster, suggesting rapid electron transfer (and thus a high yield). The ratio of the lifetimes also does not take into account the competing kinetic decay channels present for the excited state. The correct way to estimate this yield is to extract the charge transfer rate constant (I'll call it "k2" here) from the observed rate constant in the aggregate: $k_1+k_2 = 1/(134 \text{ ps})$, where k_1 is the rate of monomer decay ($k_1 = 1/1422 \text{ ps}$), using their values, and then computing the efficiency of CS using that value: the estimated yield of charge transfer is $k_2/(k_1+k_2) \sim 90\%$. The authors should revise their estimate and discuss the discrepancy between this value and the ~5% efficiency of H₂ generation. Large discrepancies are not uncommon, as surface area limitations and charge recombination pathways in the aggregates all lead to kinetic competition with catalysis, but the current discussion is insufficient and the numbers are not obtained in a rigorous way.

Response 1: We are very grateful to the reviewers for giving such comprehensive guidance. We have corrected the calculations and discussion in the relevant sections. As shown in Supplementary Notes 2:

The observed rate constant in PTA nanosheet is: $1/\tau = k_1 + k_2$

Where τ is the photogenerated electron lifetime from TAS data, 134.2 ps.

k_1 is the rate of monomer decay, $k_1 = 1/\tau_1$, τ_1 is the stimulated emission (SE) delay from TAS data, 1421.9 ps. k_2 is the charge transfer rate constant, it is further calculated that $k_2 = 0.006748$.

And then computing the efficiency carrier production (Q):

$$Q = \frac{k_2}{k_1 + k_2} = 90.6\% \quad (4)$$

As shown in Figure 2c in manuscript, the apparent quantum efficiency of hydrogen production of the PTA nanosheets at 400 nm excitation was 4.8%, which is lower than the estimated carriers generation rate (90.6%). The significant difference between the apparent quantum efficiency of hydrogen production and the carrier hydrogen production rate can be explained as follows:

On the one hand, the carriers produced were not fully used for hydrogen production, which may be attributed to processes such as carriers are used to generate reactive oxygen species. As shown in the electron paramagnetic resonance (EPR) measurement (Supplementary Fig.47), where the photogenerated electrons of PTA are trapped by dissolved oxygen to form $\cdot O_2^-$ in an air atmosphere. It further reacts with H^+ to form $\cdot OH$.

On the other hand, in general, after photoexcitation of a semiconductor to generate photogenerated excitons or carriers, these photoexcited species need to migrate or diffuse to the reaction sites to drive the reaction. However, for excitons produced in organic semiconductors, the typically low exciton migration distances limit the participation of excitons in catalytic reactions. For carriers, the average free range of electrons or holes can make them collide and annihilate or complex during migration, limiting carriers' participation in catalytic reactions. These limitations are caused by the morphological characteristics of semiconductor catalysts. For the PTA nanosheets studied in this paper, photogenerated excitons and carriers are produced in excited molecules by light. However, typical water reduction reactions only occur at the edges of the nanosheets, which leads to surface area limitations on the apparent quantum efficiency of the actual hydrogen production.

Supplementary Figure 47. EPR test of the hydroxyl radical ($\cdot\text{OH}$) and superoxide radical ($\cdot\text{O}_2^-$) signals captured by DMPO (0.1 M) in air atmosphere. It shows that $\cdot\text{OH}$ and $\cdot\text{O}_2^-$ are produced by PTA nanosheets after light exposure in an air atmosphere.

Comment 2. The addition of the mid-IR TAS data is quite useful and provides a better, more spectrally specific probe of the free carriers. Perhaps I missed them among the other data in the Supplemental Materials, but I did not see the actual mid-IR TA spectra anywhere in the manuscript or SM. The authors should provide the full spectrally resolved mid-IR TAS data similar to how they have provided the data for the visible and NIR ranges.

Response 2: Thanks for your suggestion that we have add the mid-IR TAS data in Supplementary.

Supplementary Figure 35. Mid-IR TAS of (a) PTCDA (b) PTA (c) PTA in $AgNO_3$ (d) PTA in ascorbic acid at different time delays after nanosecond laser excitation at 400 nm (in the form of coating film).

Comment 3. Similarly, there are no experimental details given for their mid-IR TAS measurements or apparatus beyond the sample preparation, which makes comparisons of the experimental conditions difficult. Full experimental details should be given in the Supplemental Materials.

Response 3: Thank you for careful suggestion. We have added more mid-IR TAS measurements or apparatus details for completeness.

Description in manuscript: Specifically, in the mid-infrared region, transient absorption spectra were acquired with the following experimental setup. The output of an 800 nm centered pulsed laser with a duration of 35 fs (FWHM) at a repetition rate of 1 kHz from an amplified Ti:Sapphire laser (Spitfire Ace, Spectra Physics) was split into two beams by a 800 nm beam splitter. One of the beams passed through a delay line was employed as the excitation beam. The other beam was used to generate a broadband mid-IR source as the mid-IR probe via four-wave mixing through filamentation in air. The fundamental beam first passed through a short delay line such that its time delay could be finely adjusted, then passed through a half-wave plate that rotated its light polarization by 90°, and finally recombined with the second harmonic beam by another dichroic mirror. Transmitted mid-IR light was collected, sent into an imaging spectrometer (iHR 320, HORIBA Jobin Yvon) and acquired by a 64-channel MCT array detector of Femtosecond Pulse Acquisition System (FPAS-0144, Infrared Systems Development). The excitation wavelength was 400 nm and with an excitation energy of 5 uJ/cm².

Reviewer #2 (Remarks to the Author):

The authors have well addressed all comments. The paper can be accepted as is.

Response: We are very thankful for the Reviewer's positive evaluation of our work.

Reviewer #3 (Remarks to the Author):

General Comment. In the revised manuscript, the authors tried the best to increase the quality by adding more experiments and giving more discussions according to the comments from the referees. I recommend acceptance of this manuscript for publication on Natural Communications. The following points need to be further considered and clarified.

Response: We are very thankful for the Reviewer's positive evaluation of our work and valuable advices that improve the quality of our manuscript. We have carefully considered all the questions that were raised. The Response to each question is described below.

Comment 1. About the exciton migration in the nanosheet, the authors just give some references (40-42) to address the advantage of the ultrathin crystal. Is it possible to give some measurements about the exciton migration length? In nanocrystal, some time very long migration distance could be observed. Indeed, in recent reports on nonfullerene electron acceptors, the typical exciton migration distance is about 50nm. So, I am wondering the excitons might migrate along the pi-pi stacking direction, since the CT coupling is important in these nanosheets.

Response 1: Thank the Reviewer for pointing out this important question, we have given the exciton migration length based on the mobility and lifetime of free carriers.

The diffusion length can be calculated by incorporating the following two equations:

$$L = \sqrt{D\tau}$$

$$\mu = \frac{e}{k_B T} D$$

where L is the diffusion length, μ denotes the carrier mobility, D represents the diffusion coefficient, τ defines the carrier lifetime and e is the elementary charge. k_B is the Boltzmann constant.

where the carrier mobility was obtained from the SCLC test ($3.2 \text{ cm}^2 \text{ V}^{-1} \text{ s}^{-1}$) and the carrier lifetime was obtained from the transient absorption spectrum (1421.9 ps). The diffusion length of the exciton was calculated to be 108.2 nm. Compared to conventional organic electron acceptors, the exciton migration distance in PTA is significantly prolonged due to the coupling between the highly ordered π -electron clouds.

Comment 2. In the introduction section, when the perylene serial materials was discussed as catalyst for hydrogen evolution, it is better to give a proper description on the strategy to use low energy sunlight since it represents one of the important progresses in this field (doi.org/10.1002/anie.202001231).

Response 2: Thank you for your introduction to these wonderful research work. According to your suggestion, we properly cite this article in introduction as:

The perylene plane series materials have become a kind of promising organic photocatalyst due to the π - π stacking electron migration channels^{20,21}, cost effectiveness and the broaden light absorption²², especially for low-energy solar energy²³.

23 Xu, Y. *et al.* Consecutive Charging of a Perylene Bisimide Dye by Multistep Low-Energy Solar-Light-Induced Electron Transfer Towards H_2 Evolution. *Angew. Chem. Int. Edit.* **59**, 10363-10367, (2020).

Comment 3. In Figure 1e left, the thickness of one layer is show as 2 nm (from 0.75 to 2.75 nm); however, the description in page 6 line 175 is 1.5 nm observed by AFM. To my knowledge, the distance between O-O atoms in PTCDA is about 1nm. So, please check the molecular size in the optimized molecular structure, and give a more accurate presentation in Figure 1e.

Response 3: Thank you for your carefully suggestion. We examined the optimized PTA molecular structure with a single molecule thickness of 1.3 nm. The thickness of the PTA nanosheet tested by AFM is 1.5 nm. This small difference is attributed to the distance between the PTA molecule and the tested substrate HOPG. Our representation at this point is fuzzier than expected, and we have added a more accurate representation in Figure 1e.

Fig. 2. Photocatalytic hydrogen evolution. (a) Photocatalytic HER under full spectrum ($\lambda \geq 300\text{nm}$, light intensity: $\sim 530 \text{ mW cm}^{-2}$), visible light ($\lambda \geq 420\text{nm}$, light intensity: $\sim 450 \text{ mW cm}^{-2}$) and AM 1.5G, (100 mW cm^{-2}) of PTA. (b) AM 1.5G (100 mW cm^{-2}) HER performance of PTA (2 mg) loaded on nonwoven fabrics, inset: Optical image of PTA loaded on nonwoven devices. (All error bars are determined from three independent experiments). (c) The wavelength-dependent AQE for photocatalytic HER over PTA. (d) The stable photocatalytic HER performance of PTA under full spectrum. Reaction conditions: 7.0 mg photocatalysts, 4.6 wt% Pt as co-catalysts, 100 mL, 0.2 M ascorbic acid solution and the pH is 2.45.

Comment 4. In Figure 2a, the hydrogen evolution rates were obtained by using different light sources. It will be better to give the spectra of these light sources.

Response 4: Thank you for your comments. Based on your suggestion, we add the spectra of full-spectrum, visible light ($\lambda \geq 420$ nm), and AM 1.5G.

Supplementary Figure 58. The spectra of the different light sources.

Comment 5. In Figure 2c, the AQE is not zero in the spectral region of >550 nm. Was it in the error range? How about the AQE values in the near IR region?

Response 5. Thank you for pointing out this issue. Figure 2c demonstrates a non-zero AQE at $\lambda = 590$ nm, but the AQE is as low as 0.2%. This value is within the error range. The hydrogen production performance at $\lambda = 800$ nm in the IR region is not detected, thus the AQE in the NIR region is 0.0% (within the sensitivity of the instrument).

REVIEWERS' COMMENTS

Reviewer #1 (Remarks to the Author):

The authors have satisfactorily addressed my concerns. My only comment is that the calculated rate constants in the yield analysis should have units, but besides that, the manuscript can be accepted in its current form.

Reviewer #3 (Remarks to the Author):

The authors have significantly improved the manuscript and all my concerns have been addressed well. So, I recommend acceptance for publication as is.

Detailed Responses to Reviewers

Manuscript number: NCOMMS-21-38971C

Manuscript Type: Article

Title: Perylenetetracarboxylic Acid Nanosheets with Internal Electric Fields and Anisotropic Charge Migration for Photocatalytic Hydrogen Evolution

Correspondence Author: Yongfa Zhu

[The reviewer comments are shown in *italic*; Responses are in **blue**;]

Reviewer #1 (Remarks to the Author):

Comment. The authors have satisfactorily addressed my concerns. My only comment is that the calculated rate constants in the yield analysis should have units, but besides that, the manuscript can be accepted in its current form.

Response: We are very thankful for the Reviewer's positive evaluation of our work. And we are grateful to the reviewers for giving such a careful check. We have added units in the yield analysis section, e.g. $k_2 = 0.006748 \text{ ps}^{-1}$.

Reviewer #3 (Remarks to the Author):

Comment. The authors have significantly improved the manuscript and all my concerns have been addressed well. So, I recommend acceptance for publication as is.

Response: We are thankful for the Reviewer's helps and positive evaluation of our manuscript.